



# Impact of increased resolution on long-standing biases in HighResMIP-PRIMAVERA climate models

**Eduardo Moreno-Chamarro**[1], **Louis-Philippe Caron**[1,a], **Saskia Loosveldt Tomas**[1], **Javier Vegas-Regidor**[1],
**Oliver Gutjahr**[2,b], **Marie-Pierre Moine**[3], **Dian Putrasahan**[2], **Christopher D. Roberts**[4], **Malcolm J. Roberts**[5],
**Retish Senan**[4], **Laurent Terray**[3], **Etienne Tourigny**[1], and **Pier Luigi Vidale**[6]

[1]Barcelona Supercomputing Center (BSC), 08034, Barcelona, Spain
[2]Max Planck Institute for Meteorology, 20146 CE1 Hamburg, Germany
[3]CECI, Université de Toulouse, CERFACS/CNRS, 31100, Toulouse, France
[4]ECMWF European Centre for Medium-Range Weather Forecasts, Reading, RG2 9AX, United Kingdom
[5]Met Office, Exeter, CE2 EX1 3PB, United Kingdom
[6]NCAS-Climate, Department of Meteorology, University of Reading, Reading, RG6 6BB, United Kingdom
[a]now at: Ouranos, Montréal, H3A 1B9, Canada
[b]now at: Institut für Mereeskunde, Universität Hamburg, 20146 Hamburg, Germany

**Correspondence:** Eduardo Moreno-Chamarro (eduardo.moreno@bsc.es)

**Abstract.** We examine the influence of increased resolution on four long-standing biases using five different climate models developed within the PRIMAVERA project. The biases are the warm eastern tropical oceans, the double Intertropical Convergence Zone (ITCZ), the warm Southern Ocean, and the cold North Atlantic. Atmosphere resolution increases from ∼ 100–200 to ∼ 25–50 km, and ocean resolution increases from ∼ 1° (eddy-parametrized) to ∼ 0.25° (eddy-present). For one model, ocean resolution also reaches 1/12° (eddy-rich). The ensemble mean and individual fully coupled general circulation models and their atmosphere-only versions are compared with satellite observations and the ERA5 reanalysis over the period 1980–2014. The four studied biases appear in all the low-resolution coupled models to some extent, although the Southern Ocean warm bias is the least persistent across individual models. In the ensemble mean, increased resolution reduces the surface warm bias and the associated cloud cover and precipitation biases over the eastern tropical oceans, particularly over the tropical South Atlantic. Linked to this and to the improvement in the precipitation distribution over the western tropical Pacific, the double-ITCZ bias is also reduced with increased resolution. The Southern Ocean warm bias increases or remains unchanged at higher resolution, with small reductions in the

regional cloud cover and net cloud radiative effect biases. The North Atlantic cold bias is also reduced at higher resolution, albeit at the expense of a new warm bias that emerges in the Labrador Sea related to excessive ocean deep mixing in the region, especially in the ORCA025 ocean model. Overall, the impact of increased resolution on the surface temperature biases is model-dependent in the coupled models. In the atmosphere-only models, increased resolution leads to very modest or no reduction in the studied biases. Thus, both the coupled and atmosphere-only models still show large biases in tropical precipitation and cloud cover, and in midlatitude zonal winds at higher resolutions, with little change in their global biases for temperature, precipitation, cloud cover, and net cloud radiative effect. Our analysis finds no clear reductions in the studied biases due to the increase in atmosphere resolution up to 25–50 km, in ocean resolution up to 0.25°, or in both. Our study thus adds to evidence that further improved model physics, tuning, and even finer resolutions might be necessary.

# 1 Introduction

Climate models have biases with respect to observations, some of which have persisted over model generations with little or no improvement (e.g., Wang et al., 2014; Tian et al., 2020). These biases can undermine the credibility of climate models, contributing to uncertainties in regional climate projections (Boberg and Christensen, 2012; Maraun, 2016) and limiting their skill in predicting the climate of coming seasons and decades (e.g., Meehl et al., 2014; Exarchou et al., 2021). Assessing and reducing common model biases are therefore key topics for the climate community to address.

Increased model resolution is frequently seen as a way to improve model realism and hence reduce climate biases. Most of the global climate models taking part in the CMIP activities have a nominal resolution of about 150 km in the atmosphere and 1° in the ocean (e.g., IPCC, 2013), which ensures a reasonable trade-off between computing time and model complexity. Higher-resolution models have shown improvements in simulating the Gulf Stream position (e.g., Kirtman et al., 2012; Moreno-Chamarro et al., 2021), the Intertropical Convergence Zone (ITCZ; e.g., Doi et al., 2012; Tian et al., 2020), and the storm tracks (e.g., Hodges et al., 2011), just to mention a few examples. Haarsma et al. (2016), Hewitt et al. (2017), and M. J. Roberts et al. (2018) have extensively reviewed the benefit of high-resolution modeling.

On this basis, the Horizon2020 PRIMAVERA project (https://www.primavera-h2020.eu/, last access: TS1) was conceived to "develop a new generation of advanced and well-evaluated high-resolution global climate models, capable of simulating and predicting regional climate with unprecedented fidelity, for the benefit of governments, business and society". Such new models have shown improvements in the representation of various aspects of weather and climate variability, including blocking frequency over the Pacific and Atlantic (Schiemann et al., 2020), the distribution of precipitation over Europe (Demory et al., 2020), tropical cyclones (M. J. Roberts et al., 2020a; Vannière et al., 2020; Vidale et al., 2021; Zhang et al., 2021), air–sea interactions over the Gulf Stream (Bellucci et al., 2021), and Atlantic Ocean heat transports (M. J. Roberts et al., 2020b). In this study, we provide a systematic assessment of the impact of ocean and atmospheric resolution on mean climate (Sect. 3), focusing on the following long-standing biases: (i) the warm bias in the eastern tropical oceans, (ii) the double ITCZ, (iii) the warm Southern Ocean (SO), and (iv) the cold North Atlantic (Sects. 4 and 5). We provide a brief introduction to each bias immediately below. The models, experimental design, and observational datasets are described in Sect. 2, while the main conclusions and the discussion of the results are in Sect. 6.

## 1.1 Biases in the tropics

### 1.1.1 Upwelling regions

The first long-standing bias examined is the warm bias in the eastern tropical oceans, which affects many state-of-the-art and previous-generation climate models (Li and Xie, 2012; Xu et al., 2014a; Richter, 2015; Richter and Tokinaga, 2020). The eastern tropical oceans are characterized by intense coastal upwelling driven by the trade winds, which bring cold, nutrient-rich waters from the deep ocean to the surface and transport them several thousand kilometers offshore. Cold surface waters contrast with warmer atmospheric temperature aloft, which generates stable atmospheric conditions that favor the formation of large-scale shallow stratocumulus decks. These reflect a large fraction of the solar radiation and thereby help sustain the cold ocean surface below. This system is misrepresented in many climate models, which fail to reproduce the cold tongue of surface waters and hence exhibit a warm bias extending offshore (see, for example, bottom left panel in Fig. 1b). This bias has long been related to the underestimation of the cloud cover, which leads to warming because of excessive shortwave radiation reaching the surface (e.g., Huang et al., 2007; Hu et al., 2008). The warm bias, in turn, weakens the lower-tropospheric stability and thus hinders the formation of the stratocumulus deck, which contributes to sustaining the surface warm bias. Other mechanisms have been proposed to explain this bias, including too weak equatorial and alongshore winds weakening upwelling (e.g., Richter et al., 2012; Koseki et al., 2018; Goubanova et al., 2019; Voldoire et al., 2019a), biases in regional atmospheric moisture (Hourdin et al., 2015), too weak offshore transport by ocean mesoscale eddies, and the misrepresentation of the coastal current system (Xu et al., 2014b) or vertical mixing in the upper ocean (e.g., Hazeleger and Haarsma, 2005; Exarchou et al., 2018; Deppenmeier et al., 2020). Richer (2015) extensively reviewed all these mechanisms.

Increased horizontal (typically beyond ∼ 25–50 km) and vertical resolution in the atmosphere can reduce the warm bias due to an improved representation of coast-parallel winds and better-resolved orography, especially along the coast of west Africa (Gent et al., 2010; Milinski et al., 2016; Harlaß et al., 2018). A mesoscale-resolving oceanic resolution can also mitigate the warm bias by improving the representation of the complex coastal current system as well as the mesoscale eddy contribution to the upper-ocean heat budget and offshore transport from the upwelling regions in the Atlantic (Seo et al., 2006; Xu et al., 2014b; Small et al., 2015). However, the bias persists in some models and ocean basins, even after increasing their resolution (Jochum et al., 2005; Doi et al., 2012; Delworth et al., 2012; Milinski et al., 2016; Goubanova et al., 2019), which suggests that a refinement of model physics might still be necessary to remove it (Patricola et al., 2012; Harlaß et al., 2018). A reduction

**Geosci. Model Dev., 14, 1–21, 2021**                                    **https://doi.org/10.5194/gmd-14-1-2021**

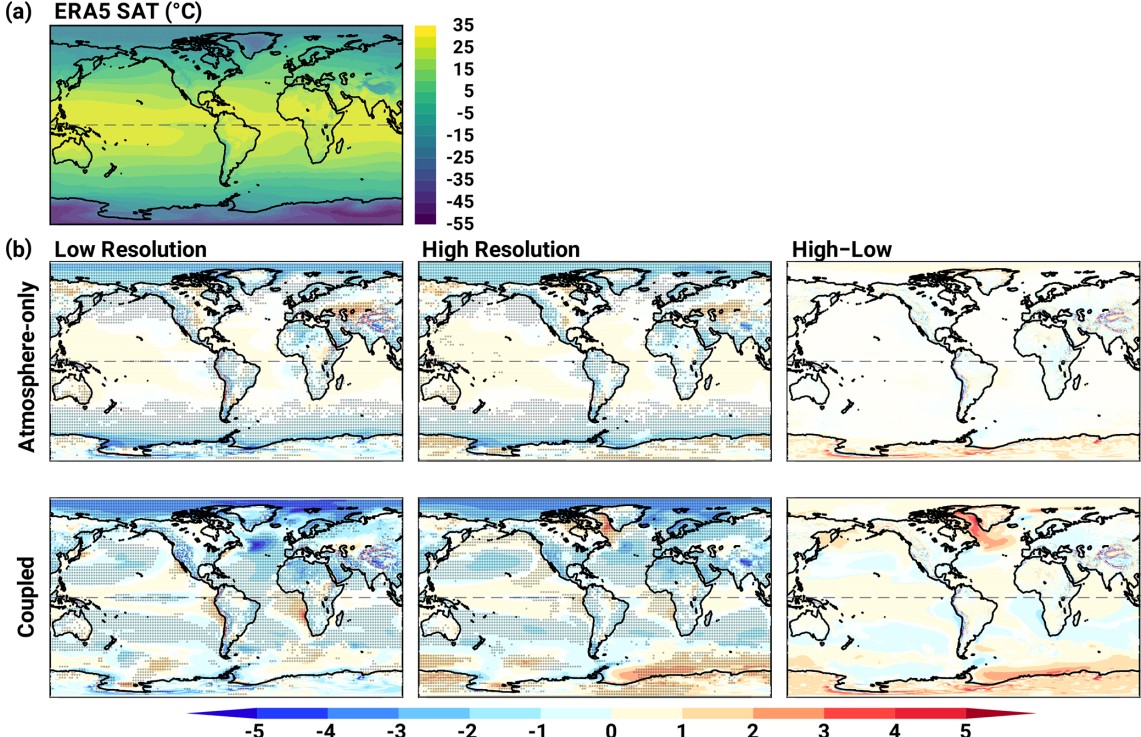

**Figure 1. (a)** ERA5 near-surface (2 m) air temperature (SAT; in °C). **(b)** Left and middle: multi-model ensemble mean bias in (2 m) SAT CE2 (in °C) in the atmosphere-only (top) and coupled (bottom) simulations at low (left) and high (middle) resolutions. Biases are with respect to ERA5 (shown in **a**). Stippling masks where at least four out of the five models agree on the anomaly sign. Right: difference between the two resolutions. In all panels non-significant anomalies at the 5 % level (based on a two-tailed Student's *t* test) are masked white. The Equator is a dashed line in all the panels.

in the temperature and cloud biases in the eastern tropical oceans might reduce current uncertainty about climate sensitivity (Andrews et al., 2019), impact precipitation biases for example over the equatorial North Atlantic (e.g., Hazeleger and Haarsma, 2005; Huang et al., 2007; Siongco et al., 2016), and enhance models' predictive skill over the tropics (Exarchou et al., 2021).

### 1.1.2 The double ITCZ

Another long-standing bias in the tropical climate in GCMs affects the representation of the ITCZ, referred to as the double ITCZ. This bias takes the form of a tropical precipitation distribution with two distinct maxima – to the north and south of the Equator – instead of a single one north of the Equator, as in observations (Fig. 2a and black line in Fig. 3; Schneider et al., 2014). The double-ITCZ problem has persisted over climate model generations (e.g., Lin, 2007; Li and Xie, 2014; Oueslati and Bellon, 2015; Zhang et al., 2015; Samanta et al., 2019; Tian and Dong, 2020); it has been related to deficiencies in the tropical or global energy budget (Hwang and Frierson, 2013; Bischoff and Schneider, 2016; Adam et al., 2016, 2018), in atmospheric deep convection (Zhang and Wang, 2006; Oueslati and Bellon, 2015; Song and Zhang,

2019), in land temperature (Zhou and Xie, 2017), and in the atmosphere–ocean coupling due to sea-surface temperature (SST) biases amplified by the wind-evaporation-surface temperature and the Bjerknes feedbacks (Lin, 2007; Li and Xie, 2014; Qin and Lin, 2018; Samanta et al., 2019). The double ITCZ commonly develops together with a cold surface bias and too weak easterlies over the equatorial western Pacific, which together lead to reduced convective precipitation aloft (Lin, 2007; Li and Xie, 2014; Oueslati and Bellon, 2015; Zhang et al., 2015; Samanta et al., 2019). The double-ITCZ bias can present distinct seasonal characteristics (Lin, 2007; Li and Xie, 2014; Oueslati and Bellon, 2015; Adam et al., 2018) – although we will focus on the annual mean in our analysis for the sake of simplicity.

Increased model resolution can alleviate the double-ITCZ bias, especially over the Atlantic when the eastern tropical warm bias is reduced (Seo et al., 2006; Delworth et al., 2012; Doi et al., 2012; Harlaß et al., 2018; Song and Zhang, 2020) and orography or mesoscale systems are better resolved in models (de Souza Custodio et al., 2017; Vannière et al., 2019; Monerie et al., 2020) and over the Pacific when tropical instability waves are explicitly resolved and extratropical, Pacific temperatures are more accurately simulated (Wengel et al., 2021). Nonetheless, strong biases in the ITCZ and trop-

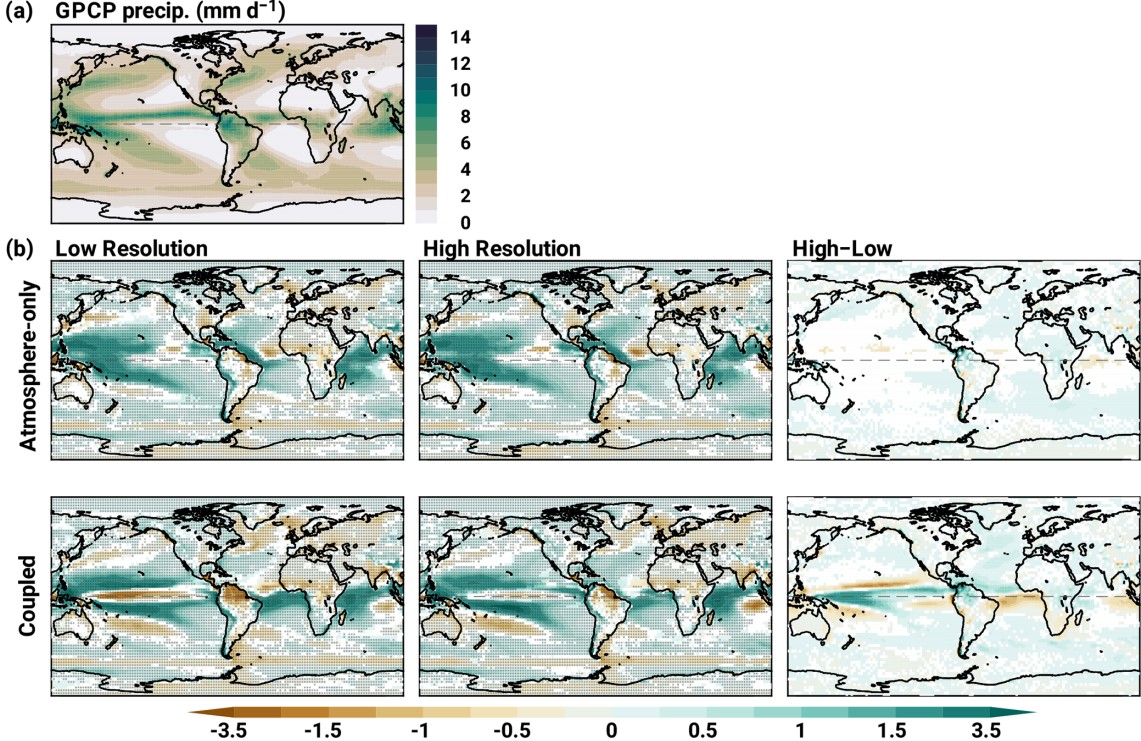

**Figure 2. (a)** GPCP precipitation rate (in mm d$^{-1}$). **(b)** Multi-model ensemble mean bias in precipitation rate (in mm d$^{-1}$) with respect to the GPCP precipitation at low and high resolution (left and middle) and differences between the two resolutions (right), as in Fig. 1.

ical precipitation still develop in higher-resolution models (Gent et al., 2010; McClean et al., 2011; Raj et al., 2019), which might further be reduced through improved convective parametrizations (Zhang et al., 2019) or the use of atmospheric convection-permitting (i.e., storm-resolving) climate models (Klocke et al., 2017).

## 1.2 Biases in middle and high latitudes

Besides biases in the tropics, climate models also present substantial biases at higher latitudes, which have also persisted across model generations. Here, we will discuss two of the best-known: the SO surface warm bias and the cold bias in the subpolar North Atlantic.

### 1.2.1 Southern Ocean

Both past and state-of-the-art climate models show a surface warm bias over extensive areas at midlatitudes and higher latitudes of the SO (see, for example, the bottom left panel in Fig. 1b; Schneider and Reusch, 2016; Beadling et al., 2020). This bias has been attributed to an excessive shortwave radiation reaching and warming the surface ocean because of the underestimation of the cloud cover (especially mixed-phase clouds) and errors in the cloud forcing (Hwang and Frierson, 2013; Bodas-Salcedo et al., 2012, 2014; Kay et al., 2016; Schneider and Reusch, 2016; Hyder et al., 2018). The ex-

tent and magnitude of these biases affects important aspects of the climate, not only over the SO but globally. Thus, for example, too warm surface temperatures result in a gross underestimation of the Antarctic sea ice by models (Beadling et al., 2020). Similarly, the associated misrepresentation of the low-level temperature gradient has been linked to an equatorward shift bias in the Southern Hemisphere (SH) upper-troposphere jet (Ceppi et al., 2012). Biases in clouds over the SO are an important uncertainty source for climate sensitivity (McCoy et al., 2015; Tan et al., 2016). The biggest reduction in the SO warm bias has recently been achieved through a more realistic representation of cloud properties over the region (Bodas-Salcedo et al., 2014; Seiki and Roh, 2020; Varma et al., 2020), which might be better characterized in higher-resolution models (Furtado and Field, 2017).

### 1.2.2 The North Atlantic

The bias in the North Atlantic surface temperature, associated with a misrepresentation of the northward turn of the Gulf Stream, is frequently reported in coupled as well as ocean-only climate models (Bryan et al., 2007; IPCC, 2013; Wang et al., 2014; Marzocchi et al., 2015). The bias is characterized by a warm anomaly off the eastern North American coast, due to a Gulf Stream separation that is too far north and a cold anomaly to the east in the central subpolar region, due to too zonal a North Atlantic Current (see, for example,

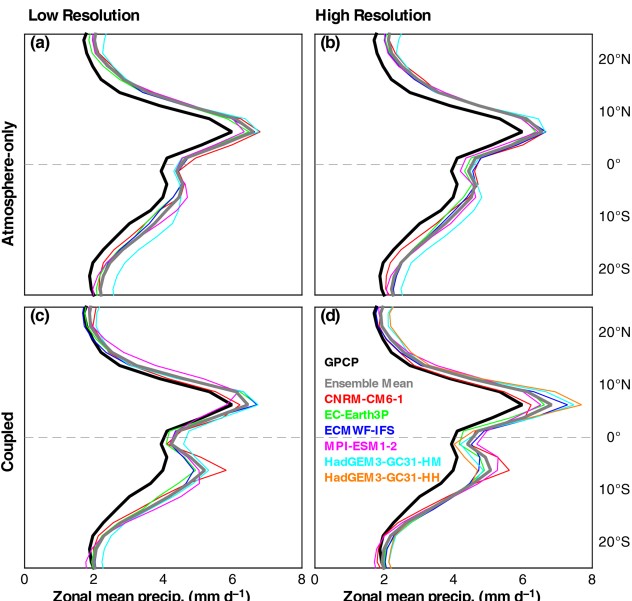

**Figure 3.** Zonally averaged precipitation rate (in $\mathrm{mm\,d^{-1}}$) in the tropics for the period 1980–2014 in the atmosphere-only **(a, b)** and coupled **(c, d)** models at low **(a, c)** and high **(b, d)** resolutions. In all the panels, the individual models are the colored, thin lines, the ensemble mean is the gray, thick line, and the GPCP dataset is the black, thick line. The HadGEM3-GC31-HH (orange line) is shown in **(d)** only.

bottom left panel in Fig. 1b). Improving the representation of the Gulf Stream and North Atlantic paths, as found in studies using ocean models at eddy-rich resolutions ($\sim 0.1$–$0.05°$; Smith et al., 2000; Bryan et al., 2007; Mertens et al., 2014), may therefore reduce the bias in North Atlantic temperatures (M. J. Roberts et al., 2019). However, ocean models at relatively high ($\sim 0.25$–$0.1°$) resolutions can still have substantial biases in subpolar North Atlantic temperature and salinity compared to $1°$- or lower-resolution models (Delworth et al., 2012; Menary et al., 2015). Instead of increased resolution, ad hoc corrections to the North Atlantic circulation and surface fluxes can also reduce the North Atlantic bias (Drews et al., 2015). The North Atlantic bias can lead to further biases in the atmospheric circulation over the entire North Atlantic and Europe (Scaife et al., 2011; Keeley et al., 2012; Lee et al., 2018) and influence the characteristics of the North Atlantic decadal variability (Menary et al., 2015); an unrealistic Gulf Stream separation can similarly affect its response to future increases in greenhouse gases (Moreno-Chamarro et al., 2021).

## 2 Experimental setup

### 2.1 Models and simulations

We compare simulations generated with five different climate models participating in the PRIMAVERA project and for which all the necessary data were publicly available on the CEDA-JASMIN platform at the time of the analysis (Table 1): CNRM-CM6-1 (Voldoire et al., 2019b), EC-Earth3P (Haarsma et al., 2020), ECMWF-IFS (C. D. Roberts et al., 2018), HadGEM3-GC31 (M. J. Roberts et al., 2019), and MPI-ESM1-2 (Gutjahr et al., 2019). Two resolutions for each model are compared (details provided in Table 1): a lower one, which in most cases features a standard $\sim 100$–$200\,\mathrm{km}$ atmosphere and an eddy-parametrized, $1°$ ocean; and a higher-resolution version with a $\sim 50\,\mathrm{km}$ atmosphere and an eddy-present, $0.25°$ ocean. For simplicity, the lower- and higher-resolution versions of each model are referred to as LR and HR, respectively. In all the models except for MPI-ESM1-2 resolution increases in both the ocean and atmosphere from LR to HR (Table 1). For the MPI-ESM1-2 only the atmosphere resolution increases, from a nominal resolution of 134 to $67\,\mathrm{km}$, both coupled to a $0.4°$ ocean. To extend the analysis and explore the benefit of an eddy-rich ocean model, we also analyze the HH coupled version of the HadGEM3-GC31 (M. J. Roberts et al., 2019), which has the same atmospheric resolution as its here-referenced HR version (41 km) but coupled to an eddy-rich, $1/12°$ ocean (Table 1). However, the results of the HadGEM3-GC31-HH model are simply discussed whenever they are relevant and are not included to compute the ensemble means, since this model has a different eddy regime compared to the other HR models.

Following the CMIP6 HighResMIP protocol, no additional tuning was applied to the HR model versions, except for a short list of parameters that explicitly change with resolution (especially for oceanic diffusion and viscosity; see, for example, Table 1 in M. J. Roberts et al., 2020b). Specific details about each model can be found in the references in Table 1. In contrast to the other models, the HR version of the ECMWF-IFS model was based on an existing configuration used operationally at ECMWF and then adapted to run at a lower resolution (C. D. Roberts et al., 2018b). We note that four of five coupled models share an ocean component based on NEMO (Nucleus for European Models of the Ocean; Madec et al., 2017): CNRM-CM6-1, EC-Earth3P, HadGEM3-GC31 use NEMO v.3.6, and ECMWF-IFS uses NEMO v.3.4, although all differ in their atmospheric and sea ice components and ocean tuning parameters (more details in the references in Table 1). Similarly, two of five models share an atmosphere component derived from the IFS (Integrated Forecasting System) of the European Centre for Medium-Range Weather Forecasts (ECMWF). Specifically, EC-Earth3P uses IFS cycle 36r4 and ECMWF-IFS uses IFS cycle 43r1. This similarity in the heritage of model configu-

**Table 1.** Model names, horizontal resolution, and vertical levels in the atmosphere and ocean along with reference papers.

| Model | CNRM-CM6-1 | | EC-Earth3P | | ECMWF-IFS | | HadGEM3-GC31 | | | MPI-ESM1-2 | |
|---|---|---|---|---|---|---|---|---|---|---|---|
| Resolution name | LR | HR | LR | HR | LR | HR | LL[2] | HM[2] | HH[2] | HR | XR |
| Atmosphere nominal resolution (km)[1] | 207 | 75 | 107 | 54 | 80 | 40 | 168 | 32 | 32 | 134 | 67 |
| Vertical levels (top level) | 91 (0.01 hPa) | 91 (0.01 hPa) | 91 (0.01 hPa) | 91 (0.01 hPa) | 91 (0.01 hPa) | 91 (0.01 hPa) | 85 (85 km) | 85 (85 km) | 85 (85 km) | 95 (0.01 hPa) | 95 (0.01 hPa) |
| Ocean resolution (°; km) | 1° (100) | 0.25° (25) | 1° (100) | 0.25° (25) | 1° (100) | 0.25° (25) | 1° (100) | 0.25° (25) | 0.08° (8) | 0.4° (50) | 0.4° (50) |
| Vertical levels | 75 | 75 | 75 | 75 | 75 | 75 | 75 | 75 | 75 | 40 | 40 |
| Reference | Voldoire et al. (2019b) | | Haarsma et al. (2020) | | C. D. Roberts et al. (2018b) | | M. J. Roberts et al. (2019) | | | Gutjahr et al. (2019) | |

[1] Calculated as the area-weighted mean grid box diagonal in Klaver et al. (2020). [2] The LL and HH configurations refer to the coupled model versions. The equivalent AMIP resolutions are the LM and HM, respectively, with the same low- (L) or high-resolution (H) atmosphere forced by a medium-resolution (M) SST field (M. J. Roberts et al., 2019).

rations might lead to similar biases across the ensemble, and thus our results on the impact of resolution may not generalize to all coupled modeling systems.

All simulations follow the HighResMIP experimental design (Haarsma et al., 2016). The experiments consist of (i) atmosphere-only simulations (highresSST-present), which are forced by daily, 0.25° SST, and sea ice concentration from the Hadley Center Global Ice and Sea Surface Temperature (HadISST.2.2.0; Kennedy et al., 2017), and (ii) coupled historical runs (hist-1950), which are forced by time-varying external forcing starting from a 50-year control spin-up that uses fixed 1950s forcing. Both the atmosphere-only and coupled experiments cover the period 1950–2014 – although here we focus mainly on the 1980–2014 period (see below). Comparing atmosphere-only and fully coupled climate models allows isolating the biases arising from atmosphere–ocean interactions.

Model simulation output is obtained from the Earth System Grid Federation (ESGF) nodes: CNRM-CM6-1 (Voldoire, 2019a, b), EC-Earth3P (EC-Earth, 2018, 2019), ECMWF-IFS (C. D. Roberts et al., 2017, 2018a), HadGEM3-GC31 (M. J. Roberts, 2017; Coward and Roberts, 2018; Schiemann et al., 2019), and MPI-ESM1-2 (von Storch et al., 2018a, b).

## 2.2 Observations and reanalysis

The climate models are compared against a suite of observational and reanalysis products. These include near-surface air temperature (SAT) and tropospheric zonal winds from the ERA5 reanalysis (Hersbach et al., 2020), precipitation rate from the version-2 GPCP dataset (Adler et al., 2003), cloud cover from the version-3 ESA Cloud_cci dataset (ESA CCI-CLOUD; Stengel et al., 2020), and net cloud radiative effect computed from the CERES-EBAF dataset (Kato et al., 2018; Loeb et al., 2018). The net cloud radiative effect is computed as the difference between the top-of-the-atmosphere upward net flux and the clear-sky component; it represents the net effect of clouds on the radiation budget at the top of the atmosphere, with negative mean values for cloud-induced cooling, and vice versa (Fig. 5a). Biases in SAT and zonal winds with respect to the ERA-Interim reanalysis (Dee et al., 2011) are very similar to those with respect to ERA5 (not shown). Similarly, biases in SST (not shown) are very similar to those in SAT, which suggests SAT biases are dominated by the SST ones over the ocean. The periods of comparison between models and observations are adapted to maximize observations' availability until the last simulated year (i.e., 2014). These periods are 1980–2014 for ERA5 and GPCP, 1982–2014 for ESA CCI-CLOUD, and 2001–2014 for CERES-EBAF. Biases are computed by adapting the ESMValTool (Eyring et al., 2020) recipe "recipe_perfmetrics_CMIP5.yml" (https://docs.esmvaltool.org/en/latest/recipes/recipe_perfmetrics.html, last access: TS2; Gleckler et al., 2008) to analyze the PRIMAVERA

models. The statistical significance of the differences between models or the ensemble means and the observations is calculated for each variable based on a two-tailed Student's $t$ test at the 5 % level, in which the null hypothesis is that the two samples (model and observations) have the same mean over the above-mentioned periods, assuming the two samples have different variances (von Storch and Zweirs, 1999). The associated non-significant values are masked in white in Figs. 1, 2, 4, 5, 6, and all the figures in the Supplement. An additional test is applied in Figs. 1, 2, 4, 5, and 6 (shows as stippling) to measure the agreement in the difference's sign of the ensemble members with respect to observations.

For the global biases and each regional bias (upwelling regions, double ITCZ, SO, and North Atlantic) we compute the mean bias and the root-mean squared deviation (RMSD; Tables 2 and S1–S4 in the Supplement). The areas where these metrics are computed are shown in Fig. S1 in the Supplement and, for the tropical upwelling regions over the SH Pacific and Atlantic, are between 105–70° W for the Pacific and 30° W–15° E for the Atlantic, both between 0–30° S, between 100–150° W and 0–30° S for the Pacific ITCZ (as in Tian and Dong, 2020), between 0–360° E and 50–70° S for the SO, and between 80–10° W and 35–65° N for the North Atlantic.

## 3 Global biases

Table 2 summarizes the values of the global RMSD and bias of four key variables: SAT, precipitation, cloud cover, and net cloud radiative effect. These variables are chosen to assess the different regional biases discussed in Sects. 4 and 5. On average, the ensemble presents a too cold, wet, and slightly cloudy climate, with excessive radiative cooling from clouds compared to observations. The coupled and atmosphere-only model versions present similar global biases at both resolutions for all variables except for SAT, for which biases are smaller in the atmosphere-only runs – consistent with these being forced by observed SSTs. In terms of RMSD, the ensemble mean presents some of the smallest values, likely because of error compensation among members.

In contrast to the ensemble mean, the EC-Earth3P and MPI-ESM1-2 coupled models are globally warmer compared to observations, mostly due to excessively warm SO/Antarctica and tropics, respectively (Table 2 and Fig. S3). Similarly, only the MPI-ESM1-2 models are insufficiently cloudy compared to observations (Table 1), which is connected to their strong biases over the tropics and subtropics (Figs. S6 and S7). The EC-Earth models are the only ones that consistently show a positive radiative forcing bias due to clouds, related to a widespread cloud overestimation over the SO (Figs. S8 and S9). Across the ensemble, the atmosphere-only and coupled CNRM-CM6-1 models show the largest RMSD values, particularly in cloud cover and net

cloud radiative effect (Table 2), whose biases are dominated by those over the tropics and high latitudes (Figs. S6–S9). This contrasts with their relatively low global mean biases, a clear sign of large error compensation between regions. The HadGEM3-GC31 and MPI-ESM1-2 models both have large global mean biases in cloud cover (respectively, excessively cloudy especially in the tropics and deficiently cloudy especially in the subtropics and midlatitudes; Fig. S7); however, these models have the smallest biases in net cloud radiative effect. These results highlight important differences across models within the ensemble. Compared to previous generation CMIP5 models, the global bias in net cloud radiative effect is lower in all the coupled models (Table 2; cf. Table 1 in Calisto et al., 2014).

The increase in resolution from LR to HR has, on average, a mixed effect on the global biases (Table 2). The temperature and net cloud radiative effect biases are reduced particularly in the coupled models, related to improvements in the eastern tropical oceans (Sect. 4) and North Atlantic (Sect. 5) mostly in the coupled versions of the HadGEM3-GC31 and ECMWF-IFS models. The precipitation and cloud cover biases increase with increased resolution, especially the cloud excess in the CNRM-CM6-1 and HadGEM3-GC31 models. This increase in global precipitation biases at higher resolution is consistent with previous literature (Vannière et al., 2020). In most cases, nonetheless, increased resolution has a small impact on the global biases. Since the study of global biases hides large regional differences, we discuss these in the following sections.

## 4 Biases in the tropics

### 4.1 Upwelling regions

Only the coupled configurations show a distinct warm bias in the eastern tropical oceans of a magnitude of up to 2–3 °C (Fig. 1) and of about 0.5 °C on average (Table S1). This bias is absent in the atmosphere-only models, as these are forced by observed SSTs (Fig. 1). At LR, the bias extends over the eastern tropical South Atlantic and South Pacific from the coast equatorward. In the Northern Hemisphere (NH) the warm bias is less evident in the models: off the Californian coast, only the CNRM-CM6-1, EC-Earth3P, and MPI-ESM1-2 models show a distinct warm bias (Fig. S3), whereas off the northwest Africa, most models present a cold bias instead – likely the result of the strong cold bias over the subpolar region (discussed in Sect. 5.2).

Increased resolution leads to a reduction in the bias over the SH ocean basins of up to about 1 °C (Fig. 1) and of about 0.3 °C on average in the ensemble mean (Table S1). The warm bias is largely reduced in both HadGEM3-GC31 HR models, although using an eddy-rich ocean model (HH) leads to no further reduction compared to the eddy-present ocean (HM) for the same ∼ 50 km atmosphere resolution (Fig. S3).

**Table 2.** TS3 Root-mean-square deviation (RMSD) and mean bias (Bias) for the variables in Figs. 1, 2, 4, and 5 in the atmosphere-only (Atm.) and coupled (Coup.) models at LR and HR, including the eddy-rich coupled model HadGEM3-GC31-HH. For each variable, the white to yellow shading reflects the RMSD gradient between its minimum (white) and maximum (yellow) values. The red to blue shading, which is centered around the zero value, represents the bias values, blue meaning an excessively cold and wet model with negative biases in the cloud cover and net cloud radiative effect compared to observations (and vice versa).

| | | SAT (°C) | | | | Precipitation (mm d⁻¹) | | | | Cloud Cover (%) | | | | Net cloud radiative effect (Wm⁻²) | | | |
|---|---|---|---|---|---|---|---|---|---|---|---|---|---|---|---|---|---|
| | | RMSD | | Bias | | RMSD | | Bias | | RMSD | | Bias | | RMSD | | Bias | |
| | | Atm. | Coup. | Atm. | Coup. | Atm. | Coup. | Atm. | Coup. | Atm. | Coup. | Atm. | Coup. | Atm. | Coup. | Atm. | Coup. |
| **Ensemble Mean** | LR | 0.74 | 1.14 | -0.30 | -0.66 | 0.85 | 0.97 | 0.25 | 0.21 | 9.42 | 9.57 | 0.18 | 0.46 | 7.03 | 7.54 | -1.52 | -1.68 |
| | HR | 0.76 | 1.13 | -0.25 | -0.51 | 0.87 | 0.93 | 0.30 | 0.25 | 9.02 | 9.31 | 0.59 | 0.86 | 6.64 | 6.98 | -1.27 | -1.27 |
| **CNRM-CM6** | | 1.30 | 1.86 | -0.60 | -1.11 | 1.19 | 1.19 | 0.27 | 0.21 | 13.47 | 13.53 | 0.20 | -0.44 | 12.33 | 12.45 | -5.45 | -5.09 |
| | | 1.12 | 2.08 | -0.42 | -1.38 | 1.24 | 1.20 | 0.33 | 0.21 | 13.74 | 13.64 | 3.14 | 2.71 | 11.27 | 11.73 | -4.43 | -3.95 |
| **EC-Earth3P** | | 0.87 | 1.16 | -0.45 | 0.24 | 0.76 | 1.00 | 0.14 | 0.17 | 8.61 | 9.13 | 0.55 | 0.02 | 8.88 | 9.48 | 1.76 | 2.65 |
| | | 0.99 | 1.31 | -0.51 | -0.15 | 0.87 | 0.93 | 0.21 | 0.22 | 7.89 | 8.34 | -0.55 | -0.74 | 8.99 | 9.35 | 0.27 | 1.11 |
| **ECMWF-IFS** | | 0.90 | 2.29 | -0.58 | -1.39 | 0.74 | 1.05 | 0.19 | 0.13 | 7.99 | 8.96 | 0.45 | 1.23 | 9.34 | 10.51 | -2.85 | -3.18 |
| | | 0.98 | 1.57 | -0.58 | -0.28 | 0.79 | 0.98 | 0.26 | 0.26 | 7.70 | 8.55 | -1.02 | -0.83 | 8.87 | 9.33 | -2.42 | -2.36 |
| **HadGEM3-GC31** | LR | 1.15 | 1.90 | -0.16 | -1.22 | 1.27 | 1.15 | 0.41 | 0.34 | 12.49 | 12.50 | 4.33 | 5.84 | 7.66 | 7.87 | -0.97 | -1.67 |
| | HR (HM) | | 1.33 | | -0.66 | | 1.19 | | 0.38 | | 13.73 | | 8.57 | | 6.97 | | -0.44 |
| | | 0.90 | | -0.06 | | 1.20 | | 0.46 | | 13.18 | | 7.53 | | 6.83 | | -0.21 | |
| | HR (HH) | | 1.23 | | -0.64 | | 1.19 | | 0.38 | | 13.69 | | 8.57 | | 6.96 | | -0.38 |
| **MPI-ESM1-2** | | 1.15 | 1.33 | 0.29 | 0.17 | 1.08 | 1.28 | 0.23 | 0.21 | 11.15 | 11.26 | -4.65 | -4.36 | 8.36 | 8.51 | -0.10 | -1.08 |
| | | 1.27 | 1.40 | 0.35 | -0.07 | 1.10 | 1.26 | 0.24 | 0.20 | 11.93 | 11.64 | -6.17 | -5.51 | 8.02 | 7.80 | 0.42 | -0.71 |

For this model and bias in particular, the increase in atmosphere resolution from a ∼ 200 to a ∼ 50 km model seems to be more beneficial than the increase in the ocean from ∼ 100 to ∼ 8 km (M. J. Roberts et al., 2019).

As with many previous-generation GCMs, the surface warm bias is associated with an underestimation of the cloud cover of up to 10 %–20 % (Fig. 4) and of about 7 % on average (Table S1) over the eastern subtropical ocean in the LR ensemble. The shape and magnitude of the cloud cover bias are similar in the atmosphere-only and coupled models, which points to deficiencies in the atmosphere models as the root cause. The CNRM-CM6-1 LR model shows the largest amplitude in the cloud cover bias of about 20 % on average (Table S1) and locally above 30 % (Figs. S6 and S7), followed by the MPI-ESM1-2 LR model, with a mean bias of about 17 % (Table S2); cloud cover biases over the upwelling regions show nearly half the amplitude in the EC-Earth3P, ECMWF-IFS, and HadGEM3-GC31 LR models (Table S2 and Figs. S6 and S7). Although the cloud cover bias persists into the atmosphere-only and coupled HR models, it is re-duced by about 10 % right along the South American western coast (Fig. 4). Increased resolution reduces the cloud cover bias over the eastern South Pacific and Atlantic in the MPI-ESM1-2 and HadGEM3-GC31 coupled models compared to their atmosphere-only versions (Figs. S6 and S7). This high-lights the importance of reducing the surface warm bias un-derneath and an improved atmosphere–ocean coupling.

The temperature and cloud biases can be connected through the bias in the net cloud radiative effect (Fig. 5), which is positive (10–20 W m⁻²) along the western coasts of the subtropical South Atlantic and North and South Pa-cific in the ensemble mean. The bias, which is dominated by the shortwave component (not shown), points to an exces-sive radiative surface warming linked to cloud cover deficit (Fig. 4). Increased resolution reduces the bias in the net cloud radiative effect by about 3 W m⁻² on average (Table S1) and by up to 10–15 W m⁻² locally in the ensemble mean (Fig. 5). This is largely because of the contributions of the HadGEM3-GC31 and MPI-ESM1-2 models, especially in their coupled configuration and, to a smaller degree, in the

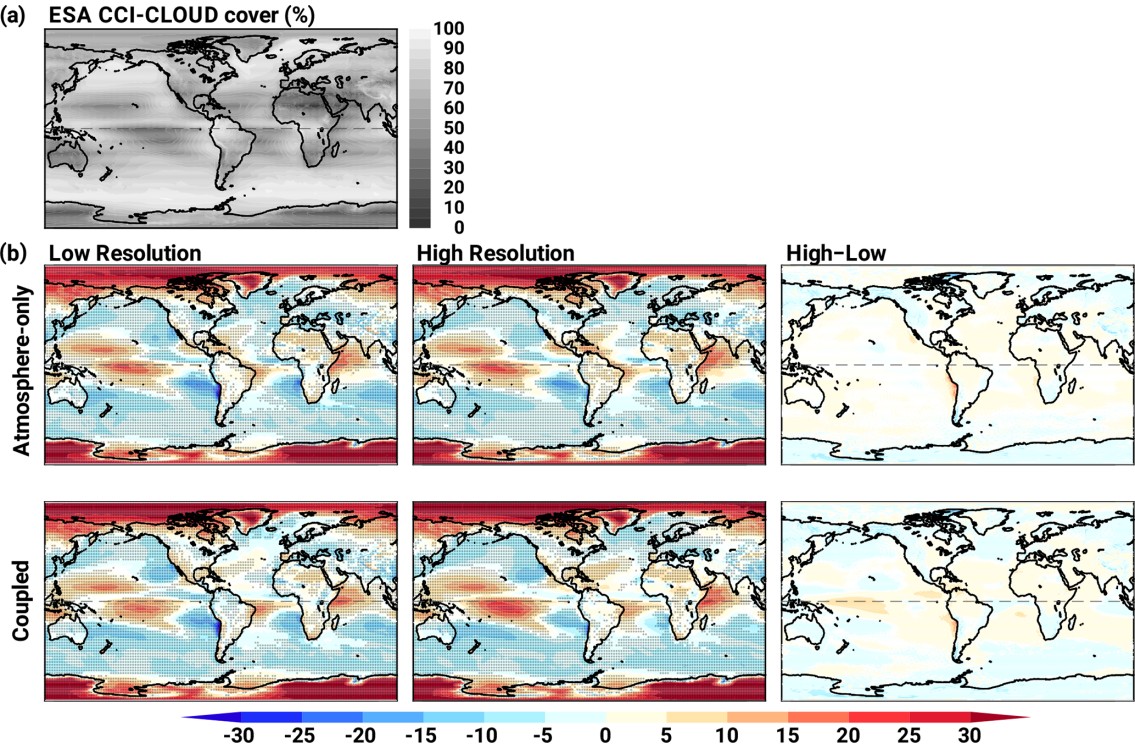

**Figure 4. (a)** ESA CCI-CLOUD cover (in %). **(b)** Multi-model ensemble mean bias in net cloud cover (in %) with respect to ESA CCI-CLOUD at low and high resolution (left and middle) and differences between the two resolutions (right), as in Fig. 1.

EC-Earth3P and ECMWF-IFS models, with local reductions of about $5\,\mathrm{W\,m^{-2}}$ (Figs. S8 and S9) as a result of the reduction in the surface warm and cloud cover biases discussed above. Contrasting with the other ensemble member, both the atmosphere-only versions and the HR coupled version of the MPI-ESM1-2 model show a negative bias in the net cloud radiative effect right along the African and South American coasts over the upwelling areas (Figs. S8 and S9), linked to a slight cloud overestimation (Figs. S6 and S7).

## 4.2 The double ITCZ

The LR models suffer from large biases in tropical precipitation (Fig. 2). These biases are similar in extent and magnitude to previous and contemporary models (CMIP3/5/6; cf. Fig. 2 in Tian and Dong, 2020). On average, the double ITCZ emerges over the Pacific basin in the coupled models (Fig. 2), where the bias presents the characteristic pattern with precipitation deficit over the Equator and excess on the northern and southern flanks by about $\mp 2\,\mathrm{mm\,d^{-1}}$ on average, respectively. This pattern can be identified in all the LR coupled models, except for CNRM-CM6-1, in which the precipitation excess is predominantly on the southern flank. Associated with the equatorial dry bias, a cold bias of up to $1\text{--}2\,°\mathrm{C}$ also affects the LR coupled models over the central equatorial Pacific (Fig. 1). In contrast to the Pacific, the precipitation bias over the tropical Atlantic points to a southward-shifted ITCZ,

with dry and wet biases to the north and south of the Equator, respectively, while over the Indian Ocean a wet precipitation bias extends over the western part of the basin and a dry one over the Indian subcontinent (Fig. 2). Such differences between ocean basins suggest that either different mechanisms are responsible for their biases or that each basin responds differently to the same large-scale/global biases. Together, the tropical precipitation biases lead to a precipitation excess mainly over the SH in the LR coupled models (Fig. 3). All the areas with precipitation excess show positive bias in cloud cover of up to about $10\,\%\text{--}20\,\%$ (Fig. 4).

In contrast to the LR coupled models, their atmosphere-only configurations show no clear double-ITCZ pattern (Figs. 2 and 3). In the zonal mean, in fact, the excess in precipitation is relatively constant across all the tropics in the atmosphere-only models (Fig. 3). This result suggests that the double ITCZ arises from misrepresented atmosphere–ocean coupling, consistent with previous literature pointing to simulated air–sea interactions and SST as key players in its development (Lin, 2007; Li and Xie, 2014; Oueslati and Bellon, 2015). The LR atmosphere-only models, instead, present excessively wet ($\sim 1.5\text{--}3\,\mathrm{mm\,d^{-1}}$; Fig. 2) and cloudy tropics ($\sim 10\,\%\text{--}20\,\%$; Fig. 4), particularly over the western parts of all the ocean basins. These regions are where the ocean surface temperature is the warmest, pointing to an excessively strong precipitation response to the imposed SST field. It is interesting to note that despite the different pattern in pre-

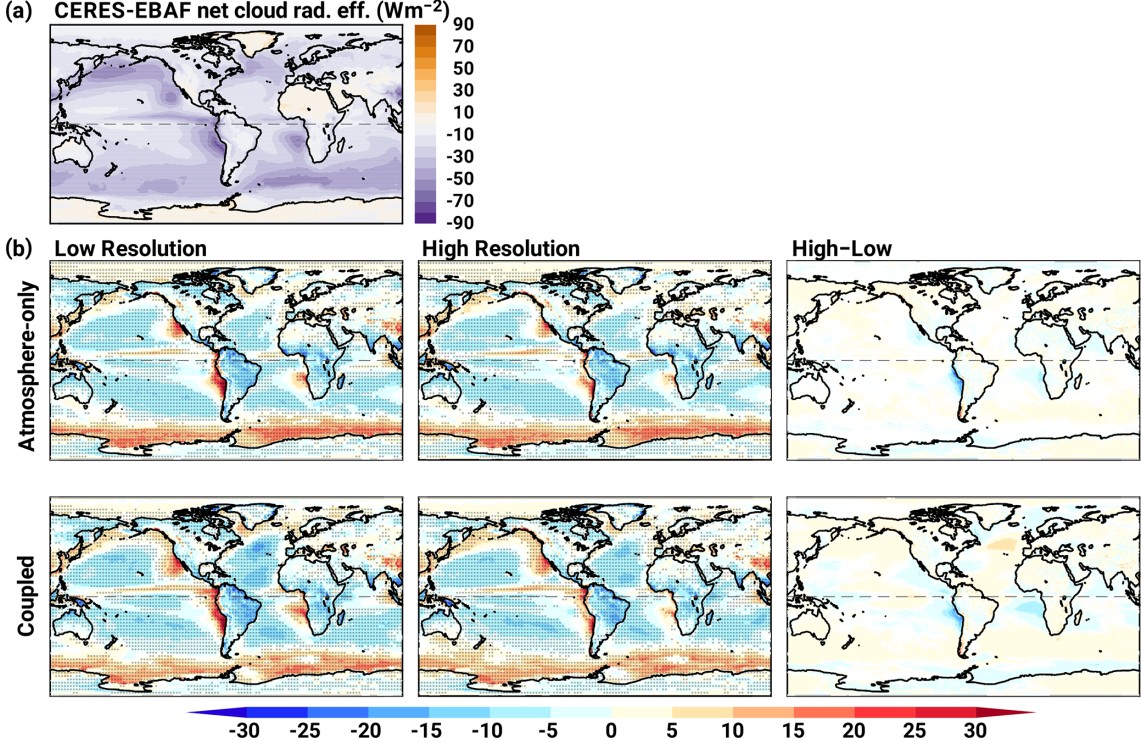

**Figure 5. (a)** CERES-EBAF net cloud radiative effect (in $W\,m^{-2}$). **(b)** Multi-model ensemble mean bias in net cloud radiative effect (in $W\,m^{-2}$) with respect to CERES-EBAF at low and high resolution (left and middle) and differences between the two resolutions (right), as in Fig. 1.

cipitation biases over the tropics between the atmosphere-only and the coupled models, their cloud biases are very similar (compare top and bottom panels in Fig. 4b and between Figs. S6 and S7). Areas with precipitation excess do not systematically present positive cloud cover biases and vice versa. This suggests that, first, errors compensate across cloud levels or types – convective cloud excess might result, for example, in a deficit in low-level, stratiform clouds – and, second, the atmosphere–ocean coupling has a subsidiary impact on the cloud bias, which most likely arises from deficiencies in the atmosphere model.

Increased model resolution reduces the tropical precipitation biases in the coupled models (Figs. 2 and 3), in agreement with previous literature (Vannière et al., 2020). In particular, the double-ITCZ bias is especially reduced over the Pacific in the ECMWF-IFS, MPI-ESM1-2, and HadGEM3-GC31 models and the southward-shifted ITCZ over the Atlantic in the HadGEM3-GC31 model (Fig. S5 and Table S2). Over these two basins, however, the bias reduction is larger for the eddy-present HadGEM3-GC31 model than for the eddy-rich one (Fig. S5 and Table S2). Over both the Pacific and Atlantic, the reduction in the tropical precipitation bias develops together with a reduction in the central equatorial Pacific cold bias of up to about 1 °C and in the eastern tropical South Atlantic warm bias (Fig. 1), in agreement with previous literature (Huang et al., 2007; Xu et al., 2014a; Siongco

et al., 2015; Song and Zhang, 2019). By contrast, cloud biases over these regions increase by about 3 % on average in the ensemble mean and locally by up to 5 %–10 % with increased resolution in the coupled models (Fig. 4) and especially in the CNRM-CM6-1, MPI-ESM1-2, and HadGEM3-GC31 models (Figs. S7 and Table S2). In most of the coupled models, increased resolution leads to modest bias reductions (overall smaller than the magnitude of the bias itself), and thus the models still exhibit large biases in precipitation and cloud cover over the tropical Pacific and Indian oceans (Figs. 2 and 4) and a clear excess in tropical precipitation (Fig. 3).

In the atmosphere-only models, bias reduction due to resolution in precipitation and clouds in the tropics is mostly negligible in the ensemble mean, and only the HadGEM3-GC31 and CNRM-CM6-1 models show a slight reduction over the western tropical North Pacific and tropical North Pacific, respectively (Figs. 2 and S4). This points to issues with the atmospheric model physics, which remained unchanged between LR and HR (Sect. 2), as the root of the precipitation and cloud cover biases over the tropics. Improvements seen in the HR coupled models therefore arise from increased resolution/improvements in the ocean, better represented coupling, or both.

## 5 Biases in middle and high latitudes

### 5.1 Southern Ocean

The SO warm bias does not appear in all the LR coupled models (Figs. 1 and S3). The EC-Earth3P and ECMWF-IFS models, which both use a combination of an IFS model and a NEMO model – albeit different versions (Sect. 2) – show a mean SAT bias of about 1 °C over the entire SO (Table S3) with local values of up to 2–3 °C (Fig. S3). By contrast, the CNRM-CM6-1, MPI-ESM1-2, and HadGEM3-GC31 models show a mean SO bias of about −1 °C, but the patterns are more mixed, with successive regional warm and cold biases that might result from a different spatial distribution in sea ice. Together with the SO warm bias, the LR coupled ensemble (and especially the CNRM-CM6-1, EC-Earth3P, and ECMWF-IFS models; Fig. S7) shows a mean underestimation of the midlatitude cloud cover by 5 %–10 % (Figs. 4, S7, and Table S3) and a positive mean bias in the net cloud radiative effect of 5–15 W m$^{-2}$ (Figs. 5, S9, and Table S3), which is dominated by the shortwave component (not shown). The MPI-ESM1-2 model shows the smallest (1 W m$^{-2}$ on average; Table S3) and least widespread bias in its net cloud radiative effect over the SO (Fig. S9), which might explain its smaller surface temperature biases (Fig. S3). In contrast to the other models, the HadGEM3-GC31 model shows a positive bias in cloud cover over the SO (Fig. S7; related to a recently introduced mixed-phase cloud parametrization; Bodas-Salcedo et al., 2019) and an overly strong net cloud radiative effect (Fig. S9); these biases contrast with its weak SO warm bias (Fig. S3) and point to some form of compensating errors (potentially due to the air–sea heat fluxes; Hyder et al., 2018; Williams et al., 2017) leading to reasonable surface temperatures. These results agree with previous studies relating the SO warm bias to the underestimation of the albedo of clouds (Bodas-Salcedo et al., 2012, 2014). The LR coupled models also present a dry bias at midlatitudes (Fig. 2). Similarly, they exhibit an equatorward shift in the upper-level jet, even in models with a relatively small SO warm bias, with too weak a zonal wind between the surface and the tropopause at around 60° S and too strong a zonal wind at upper levels ($\sim$ 200–300 hPa) to the Equator (Fig. 6), in agreement with previous studies (Ceppi et al., 2012).

Increased resolution has a mixed effect on the SO warm bias and, although it seems to increase in the ensemble mean (Fig. 1), this varies substantially across models (Fig. S3 and Table S3): the CNRM-CM6-1 model experiences a reduction in a cold bias over the Weddell Sea of up to about 4 °C; the EC-Earth3P warms along the Antarctic coast and its widespread SO warm bias persists at HR; the ECMWF-IFS model shows an increase in its temperature bias by about 1.5 °C on average and very strongly locally in the Weddell Sea by over 5 °C; the MPI-ESM1-2 shows a mean cooling over the SO of about 0.5 °C and becomes cold biased especially to the west of the Antarctic Peninsula; and the

HadGEM3-GC31 model shows a reduction in its coastal cold bias, developing instead a more widespread warm bias with local values of up to about 1–2 °C – although the cold bias over the Weddell Sea persists in the HadGEM3-GC31 eddy-rich model. In contrast to temperature, biases in cloud cover and net cloud radiative effect remain relatively unchanged between LR and HR (Figs. 4 and 5). The CNRM-CM6-1 shows a 1 % reduction in its mean cloud cover bias over the SO, while the ECMWF-IFS and MPI-ESM1-2 models show a 1 %–3 % increase over the SO (Table S3). Similarly, the ECMWF-IFS model shows a 1.5 W m$^{-2}$ mean reduction, while the MPI-ESM1-2 model shows a 4 W m$^{-2}$ mean increase in their net cloud radiative effect biases over the SO (Figs. S6–S9). Given the small reduction in the cloud cover and net cloud radiative effect biases with increased resolution, the change in the temperature bias over the SO might be related to a change in the sensitivity of the HR coupled models to the similar cloud and radiation biases or to development of further biases, for example, in the sea ice, mixed layer depth, air–sea heat fluxes, or the strength of the Antarctic Circumpolar Current (e.g., C. D. Roberts et al., 2018b). Some of these biases might, in turn, be linked to the disabling or not of the mesoscale eddy mixing at higher resolution (C. D. Roberts et al., 2018b), as discussed in Sect. 6. The dry bias over the SO remains unchanged (mean changes overall below 0.1 mm d$^{-1}$) with increased resolution (Fig. 2). In agreement with previous studies, there is no obvious linkage between the magnitude of the SO bias and the double-ITCZ bias in the LR and HR coupled models (Hawcroft et al., 2017). Increased resolution deepens the magnitude of the zonal wind bias over the SH in all the models, although it has little impact on the overall pattern (Fig. 6).

As for the atmosphere-only models, temperature biases over most of the SO are negligible both at LR and HR (Fig. 1). The LR versions of the CNRM-CM6-1, EC-Earth3P, ECMWF-IFS, and MPI-ESM1-2 models show a cold bias of up to 2–4 °C off the Antarctic coast, bias that is reduced only in the CNRM-CM6-1 by 1–2 °C at HR (Fig. S2); this coastal cold bias might reflect an issue in the response of the lower atmosphere to the imposed sea ice field, perhaps related to assumed ice/snow thickness used in the land-surface scheme to calculate skin temperature over ice. Biases in precipitation, cloud cover, and cloud radiative effect are comparatively similar to those in the coupled models and show negligible improvements with resolution as well (Figs. 2–5). It is interesting to note that all the atmosphere-only models show a rather zonally uniform positive bias in the net cloud radiative effect of 5–15 W m$^{-2}$ on average (Fig. S8 and Table S3). Biases in the SH jet in atmosphere-only models are similar but of smaller amplitude compared to those in the coupled models (Fig. 6).

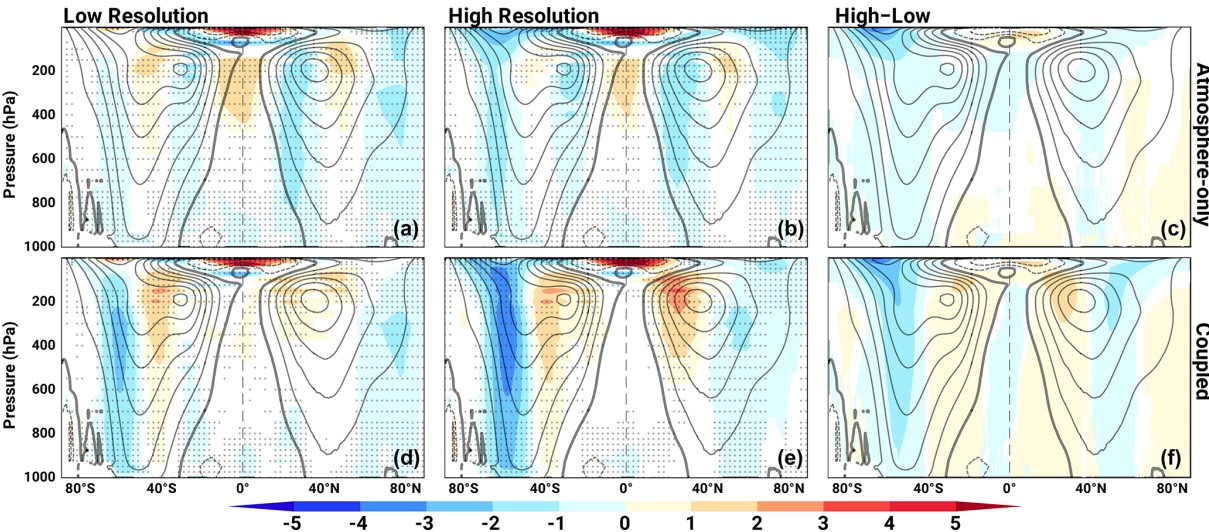

**Figure 6.** Multi-model ensemble mean bias in the zonally averaged zonal wind (in $m\,s^{-1}$) with respect to ERA5 at low and high resolution **(a, d** and **b, e)** and differences between the two resolutions **(c, f)**, as in Fig. 1. Contours represent the ERA5 climatology (every $5\,m\,s^{-1}$; negative values, for easterlies, are dashed lines, and positive values, for westerlies, are solid lines).

## 5.2   The North Atlantic

All the LR coupled models show a cold bias over the central subpolar North Atlantic and a warm one off the North American east coast, with local values of up to $-5$ and $2\,°C$, respectively, in the ensemble mean (Figs. 1 and S3). These temperature biases are absent in the atmosphere-only models, which supports the notion that these are the result of the misrepresentation of the Gulf Stream separation and path by the ocean model. The cold bias is especially strong in the ECMWF-IFS model, where anomalies colder than $-5\,°C$ cover the large areas of the subpolar North Atlantic and Nordic Seas (Fig. S3); this strong cold bias results from an unrealistically weak Atlantic meridional overturning circulation (AMOC) and related heat transport, potentially related to the lack of re-tuning compared to its HR version (see Sect. 2 and C. D. Roberts et al., 2018b). The cold bias also extends northward into Arctic latitudes in the CNRM-CM6-1 and HadGEM3-GC31 models, which points to a misrepresentation of the Arctic sea ice in addition to the Gulf Stream path and the poleward oceanic heat transport. The cold bias over the subpolar North Atlantic is accompanied by a dry bias of up to about $1\,mm\,d^{-1}$ (Fig. 2) and, in most cases, by a reduced cloud cover of up to about $10\,\%$ (Fig. 4). The cold bias might also be related to the southward-shifted jet in the NH in some models (Fig. 6) due to a southward shift in the maximum of the horizontal temperature gradient (not shown); however, the bias in the NH jet might also be related to a southward shift in the ITCZ/Hadley Circulation (especially in the Atlantic Basin; Fig. 2) and the associated intensification of the subtropical jet.

Increased model resolution reduces the magnitude of the cold bias by about $1\,°C$ on average (Table S4) and locally

by up to $2–3\,°C$ in the ensemble mean (Fig. 1). There are, however, important differences across the ensemble members (Fig. S2). The EC-Earth and CNRM-CM6-1 HR models show relatively small local reductions in the cold bias by about $0.5–1\,°C$ over the central subpolar North Atlantic. The lack of a clear improvement in these two HR models might be related to the unchanged ocean physics between the low and high resolutions (Sect. 2). The MPI-ESM1-2 shows no changes in the biases between resolutions over the subpolar North Atlantic but a strong cooling of up to about $4\,°C$ over the Nordic Seas, likely related to misrepresented local sea ice. The lack of changes in the subpolar North Atlantic biases might be because both the LR and HR MPI-ESM1-2 models use the same ocean resolution ($0.4°$; Table 2) and both present too zonal a North Atlantic Current (Müller et al., 2018). Especially remarkable are the ECMWF-IFS and HadGEM3-GC31 models, for which the cold bias is strongly reduced (Fig. S3). In the ECMWF-IFS model, this results from a much more realistic AMOC heat transport and sea ice extent in the North Atlantic compared to the LR version (C. D. Roberts et al., 2018b). In the HadGEM3-GC31, the bias is reduced thanks to the improvement in the Gulf Stream/North Atlantic path and in the northward oceanic heat transport with increased resolution (M. J. Roberts et al., 2019; Grist et al., 2021). The increase in ocean resolution from an eddy-present to an eddy-rich model leads to a more accurate Gulf Stream representation (Moreno-Chamarro et al., 2021) and a reduced warm bias near the coast (Fig. S2; M. J. Roberts et al., 2019).

On average at HR, the cold bias over the subpolar North Atlantic is replaced by a warm bias of up to about $2–3\,°C$ over the Labrador Sea (Fig. 1). The warming of the entire subpolar North Atlantic is, in fact, one of the most re-

markable differences at increased resolution in the ensemble mean. The warming is especially prevalent in the NEMO models at the 0.25° resolution, in which the warm bias is likely related to a stronger oceanic heat transport in the North Atlantic and reduced sea ice than at lower resolutions (M. J. Roberts et al., 2020b), linked to too strong an ocean deep mixing in the Labrador Sea (Koenigk et al., 2021). In the MPI-ESM1-2 models, by contrast, a warm bias is already present at LR and, together with the cold bias in the central North Atlantic bias, remains unchanged at HR (Fig. S3). It is interesting to note that these two model versions share the same ocean resolution (Table 1). These results highlight the importance of ocean resolution for the North Atlantic bias.

Changes in other biases due to resolution include a reduction in the dry bias over the subpolar North Atlantic (Fig. 2), likely related to the surface warming, and a deepening of the bias in the NH upper-troposphere jet (Fig. 6), which might be related to an intensification in eddy momentum transfer to the jet due to resolution (Willison et al., 2013) and/or to the changes in the vertical structure of the temperature bias across models. The change in the cloud cover bias in the ensemble means is relatively small – of about ±5 % over the entire North Atlantic – with no clear changes in the pattern (Figs. 3).

## 6 Discussion and conclusions

This paper examines whether increased horizontal resolution alone reduces four well-known, long-standing climate biases in five global models developed within the PRIMAVERA project. These biases are the warm eastern tropical oceans, the double ITCZ, the warm Southern Ocean (SO), and the cold North Atlantic. The analysis also considers changes in global biases. We compare atmosphere-only and fully coupled models to separate biases arising from poorly resolved atmospheric and oceanic processes alone or from atmosphere–ocean coupling. The increase in resolution in the atmosphere and ocean goes from the traditional 200–100 km grid to a 25–50 km one. The analysis also includes an eddy-rich global coupled model at an 1/12° ocean resolution. Models are compared to observations and the ERA5 reanalysis over the period 1980–2014.

All the LR coupled models suffer from the above-mentioned four key biases, as in previous and contemporary generations (CMIP3/5/6; IPCC, 2013; Wang et al., 2014; Tian and Dong, 2020). Although increased resolution contributes to reducing some of these biases, both globally and regionally, this is only in a few models and is model-dependent, for example, for surface temperature biases. In the ensemble mean, the warm eastern tropical ocean, the double ITCZ, and the cold North Atlantic biases are reduced in the coupled models at higher resolutions; by contrast, the SO warm bias increases or persists in some models, with small changes in the cloud cover and net cloud radiative effect bi-

ases aloft; finally, a new warm bias emerges in the Labrador Sea that might be related to excessive oceanic deep mixing in the coupled models using the NEMO ocean model at 1/4–1/12° resolution (Koenigk et al., 2021). Despite some improvements, large biases remain at higher resolutions, especially in precipitation and cloud cover over the tropics and in the midlatitude upper-tropospheric zonal wind, for which the benefit from resolution is rather modest. Our results are in line with previous modeling work that suggests reductions in biases due to increased resolution (e.g., Mertens et al., 2014; Harlaß et al., 2018; Monerie et al., 2020; Vannière et al., 2020) or not at all, depending on the model and region (e.g., Delworth et al., 2012; Menary et al., 2015; Raj et al., 2019; Bador et al., 2020). The emergence of a consistent warm bias in the Labrador Sea at a high resolution poses the question of what new other biases might appear at increased resolution and highlights the difficult task of removing all the model biases.

The ensemble means hide important differences across the individual models. Compared to their respective LR versions, the CNRM-CM6-1 HR model shows a modest reduction in most of its biases, although it still exhibits some of the largest biases in precipitation, cloud cover, net cloud radiative effect over the tropics, and zonal winds at SH midlatitudes among the ensemble. The EC-Earth3P HR model improves slightly in the upwelling and subpolar North Atlantic regions but still shows large biases in tropical precipitation and a widespread SO warm bias. The ECMWF-IFS HR model, the one with the finest atmospheric nominal resolution (∼ 40 km; Table 1), shows a big reduction in the North Atlantic cold bias because of a much more realistic Atlantic Ocean heat transport compared to LR and a modest bias reduction in the tropical precipitation and the eastern tropics; however, it also shows an increase in the SO warm bias and no major changes in its global cloud cover biases. The HadGEM3-GC31 HR models improve the most among the ensemble because all its biases, except for the warm SO, are reduced with increased resolution. This includes notable gains in the tropical South Atlantic upwelling region, with a bias reduction in surface temperature, cloud cover, and precipitation over the upwelling region, and in the North Atlantic. Last, the MPI-ESM1-2 HR model improves in all the regions except for the North Atlantic, where the LR and HR, both with the same ocean resolution, suffer from similar biases in the Gulf Stream path and North Atlantic temperatures. These results illustrate how strongly model-dependent the impact on the studied biases due to increased resolution can be.

When additional model configurations are available, the benefit of bias reduction from increasing ocean resolution alone can be assessed. For the ECMWF-IFS model, increased ocean resolution from 1 to 0.25° reduces the North Atlantic, Arctic, and equatorial Pacific temperature biases but increases the SO warm biases (C. D. Roberts et al., 2018b). For the HadGEM3-GC31 model, increased ocean resolution of up to an eddy-rich one (0.08°) improves the

Gulf Stream separation (M. J. Roberts et al., 2019) and representation (Moreno-Chamarro et al., 2021), although the eddy-rich resolution by itself has a modest impact on reducing surface temperature biases compared to the eddy-present (0.25°; Fig. S3 and M. J. Roberts et al., 2019). For the MPI-ESM1-2 model, the North Atlantic temperature and the Gulf Stream separation are also more realistic in an eddy-rich ocean ($\sim 0.1°$) compared to the LR and HR versions used in our study (Gutjahr et al., 2019). These results thus suggest that an eddy-rich ocean resolution might be key to reducing North Atlantic and Southern Ocean temperature biases, which is consistent with previous studies (e.g., Mertens et al., 2014; Xu et al., 2014b). Particularly important for such biases might be the treatment of the mesoscale eddy mixing at the eddy-present resolution because mesoscale eddies become smaller at higher latitudes and are therefore not fully resolved at the eddy-present resolution (0.25°). Thus, for example, while the CNRM-CM6-1 (Voldoire et al., 2019b), EC-Earth3P (Haarsma et al., 2020), and MPI-ESM1-2 (Gutjahr et al., 2019) HR models, respectively, use a Smagorinsky scheme and the Gent and Mcwilliams (1990) and K-profile parameterizations (KPP), the ECMWF-IFS (C. D. Roberts et al., 2018b) and HadGEM3-GC31 (M. J. Roberts et al., 2019) HR models switched off the Gent and Mcwilliams (1990) parametrization. Subtle differences in the model physics due to increased resolution might therefore exert a strong influence on model biases.

As for the increase in atmosphere resolution alone, it contributes to reducing the warm bias over the eastern tropical oceans in the ECMWF-IFS (C. D. Roberts et al., 2018b), HadGEM3-GC31 (M. J. Roberts et al., 2019), and MPI-ESM1-2 (this study) coupled models. Previous studies have linked a similar bias reduction to a more realistic coastal wind system (Small et al., 2015; Milinski et al., 2016). The reduction in the surface warm bias, in turn, reduces the regional precipitation and cloud cover biases aloft. However, increased atmosphere resolution alone leads to very modest bias reductions over most regions in the atmosphere-only models, which still show strong biases in tropical precipitation over the western ocean basins at HR. The atmosphere-only models also show biases in cloud cover and net cloud radiative effect over the whole tropics and in the zonal winds at midlatitudes very similar to those in the coupled models both at LR and HR.

Even though we acknowledge that our conclusions might be both model and region dependent, taken together, our analysis suggests that to remove model biases (i) a refinement of the atmosphere resolution of up to $\sim 50$ km alone might not always be sufficient and (ii) reaching eddy-rich ocean resolutions (1/12° or finer) might be needed. The increase in ocean resolution from eddy-parametrized ($\sim 100$ km) to eddy-rich ($\sim 10$ km) allows models to represent the first baroclinic Rossby radius and might therefore improve the representation of small-scale dynamical processes and then biases. In contrast, equivalent phenomena in the atmosphere

are already well resolved (the first Rossby radius at midlatitude is about 1000 km, which corresponds to the synoptic scale). Many of the challenges of reducing atmospheric model biases are related to interactions between dynamics, radiation, and parameterized (moist) physics (clouds, convection, radiation). These errors are much more difficult to address with increasing resolution as they are not obviously related to errors in grid-scale dynamics but in model physics (Kay et al., 2016; Varma et al., 2020). Increased atmospheric resolution improves the representation of weather or extremes, as found, for example, for tropical cyclones (M. J. Roberts et al., 2020a; Vannière et al., 2020; Vidale et al., 2021; Zhang et al., 2021) and blocking frequency (Schiemann et al., 2020) in PRIMAVERA models and in numerical weather prediction systems (e.g., Lean et al., 2008).

Besides increased resolution, improvements in model parametrizations and process representations, specific corrections applied to models, additional tuning, and longer spin-ups might all be essential to minimize model biases. More realistic cloud physics based on observational constraints, for example, can reduce the SO biases in the net cloud radiative effect by about $4\,\mathrm{W\,m^{-2}}$ and in the surface temperature by about $1\,°\mathrm{C}$ (Kay et al., 2016; Varma et al., 2020). Corrections to the North Atlantic Current flow and North Atlantic surface freshwater budget can suppress the cold North Atlantic bias entirely (Drews et al., 2015). Further model tuning and longer spin-ups are still to be explored. For the PRIMAVERA models considered here, no additional tuning was performed with the change in resolution, in agreement with the HighResMIP protocol (Haarsma et al., 2016). For the ECMWF-IFS model in particular, the LR version may have benefited from further tuning of the ocean component to reduce biases in the AMOC and North Atlantic SST in multi-decadal climate simulations (Fig. S2; C. D. Roberts et al., 2020a). However, in this case, it was an explicit decision to keep the ocean vertical physics as consistent as possible across configurations to ensure the LR ocean was a good proxy for the HR ocean in coupled projections at daily to seasonal timescales (C. D. Roberts et al., 2020a, b). Regarding longer spin-ups, the PRIMAVERA models considered here also followed the HighResMIP protocol, which recommended a relatively short 50-year spin-up (Haarsma et al., 2016). In the HadGEM3-GC31 LR coupled model, such a spin-up was found insufficient to stabilize its large-scale circulation and could therefore have contributed to accentuating some of its biases (M. J. Roberts et al., 2019). Testing the benefit of model re-tuning and longer spin-ups would, however, be extremely time and resource consuming if performed following traditional approaches at the highest resolutions. Further bias reduction might be gained by using new convection-permitting climate models, as computing power increases with every new model generation (Klocke et al., 2017).

To summarize, our study finds limited benefit from increased resolution alone between the traditional $\sim 100$ km

models and the ∼ 25 km ones to reduce long-standing biases, based on an ensemble of high-resolution models developed for the PRIMAVERA project. At this resolution range, increased resolution in both the atmosphere and ocean can, to some extent, reduce biases in the eastern tropical oceans, ITCZ, and North Atlantic, with further gains at an eddy-rich ocean resolution. Reductions in surface temperature biases are strongly model-dependent in the coupled models and might be subject to differences in model physics between them. In addition to further increases in resolution, we therefore propose that future efforts should also be directed toward improving model physics, for example in cloud representation, and developing innovative high-resolution model tuning approaches at higher resolutions.

*Code and data availability.* The model data used in this work are available from ESGF (https://esgf-index1.ceda.ac.uk/search/cmip6-ceda/TS4) via the references provided in Sect. 2.1. Data of ERA-5 are freely available at https://www.ecmwf.int/en/forecasts/dataset/ecmwf-reanalysis-v5 (Hersbach et al., 2020; https://doi.org/10.24381/cds.6860a573 TS5), those of GPCP at https://psl.noaa.gov/data/gridded/data.gpcp.html (Adler et al., 2003; https://doi.org/10.7289/V56971M6 TS6), those of ESA cloud cover at https://climate.esa.int/en/projects/cloud/data/ (Stengel et al., 2020; https://doi.org/10.5676/DWD/ESA_Cloud_cci/ATSR2-AATSR/V003 TS7), and those of CERES-EBAF at https://ceres.larc.nasa.gov/data/ (Kato et al., 2018; Loeb et al., 2018). Data and scripts to reproduce the figures can be obtained from https://doi.org/10.5281/zenodo.5006136 (Moreno-Chamarro, 2021).

*Supplement.* The supplement related to this article is available online at: https://doi.org/10.5194/gmd-14-1-2021-supplement.

*Author contributions.* EMC and LPC analyzed the model output. SLT and JVR assisted in using ESMValTool and the ERA5 dataset. EMC wrote the paper with input from all the authors.

*Competing interests.* The contact author has declared that neither they nor their co-authors have any competing interests.

*Acknowledgements.* This research has been supported by the Horizon2020 project PRIMAVERA (H2020 GA 641727) and IS-ENES3 (H2020 GA 824084). Eduardo Moreno-Chamarro acknowledges funding from the Spanish Science and Innovation Ministry (Ministerio de Ciencia e Innovación) via the STREAM project (PID2020-114746GB-I00) and from the ESA contract CMUG-

CCI3-TECHPROP. Etienne Tourigny has received funding from the European Union's Horizon 2020 research and innovation program under the Marie Skłodowska-Curie grant agreement no. 748750 (SPFireSD project).

*Financial support.* This research has been supported by the Horizon 2020 (PRIMAVERA (grant no. 641727), IS-ENES3 (grant no. 824084), and SPFireSD (grant no. 748750)), the European Space Agency (grant no. CMUG-CCI3-TECHPROP), and the Spanish Science and Innovation Ministry (Ministerio de Ciencia e Innovación) (STREAM (grant no. PID2020-114746GB-I00)).

*Review statement.* This paper was edited by Qiang Wang and reviewed by two anonymous referees.

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

**Remarks from the language copy-editor**

CE1  Commas removed here and in affiliation b in keeping with standard German address formatting.

CE2  Please double-check the change. I was unsure if "(2 m)" should also be removed. If so, please let us know.

**Remarks from the typesetter**

TS1  Please provide this information, day/month/year.

TS2  Please provide this information, day/month/year.

TS3  Please note that I did not find a version of Table 2 in the review which is identical to the version you provided during proofreading. Please note that the table cannot just be replaced at this stage. According to our standards, all changes in values must first be approved by the editor, as data have already been reviewed, discussed and approved. Please provide a detailed explanation for those changes that can be forwarded to the editor. Please note that this entire process will be available online after publication. Upon approval, we will make the appropriate changes. Thank you for your understanding.

TS4  Please provide a reference list entry including creators, title, repository/publisher, and date of last access.

TS5  Please provide a corresponding reference list entry for the DOI.

TS6  Please provide a corresponding reference list entry for the DOI.

TS7  Please provide a corresponding reference list entry for the DOI.

TS8  Please provide exact date.

TS9  Please provide exact date.

TS10  Please provide exact date.

TS11  Please provide exact date of last access.