# Peer review of "Impact of increased resolution on long-standing biases in HighResMIP-PRIMAVERA climate models"

_Geoscientific Model Development, 2021_

## Author Comment (AC1)

**REFEREE REPORT(S):**
**Referee: 1**

General Comments:

This manuscript presents an assessment of the HighResMIP models regarding four well-known, long-standing biases. Authors analyse simulations of 5 climate models in at least two different resolutions. Four of the five models have increased resolution in the atmospheric and oceanic components simultaneously, whereas 1 model has only increased resolution in the atmospheric component. The set of experiments consists of historical runs from 1950 to 2014 in two versions: AMIP and coupled simulations. The author's results suggest some improvements of the model performance regarding the analysed biases with increased atmospheric resolution. Yet, those improvements are not consistently found in all models. No systematic improvement is shown by increasing the resolution from eddy-parametrized into eddy-permitting ocean models. However, the only eddy-resolving ocean model presents improvements in reproducing the North Atlantic temperatures and the path of the Gulf Stream. Overall this study shows limited benefits for reducing the long-standing biases from the high resolution based on the ensemble mean of HighResMIP. This result yields a recommendation for the modelling community: in addition to model resolution, future efforts should be oriented to improve model physics.

The manuscript is well written and structured, the methodology is correctly explained, and the results are relevant. All in all, I believe that this manuscript is relevant for the climate modelling community, and it is worth to be published in GMD. In some places, the results described in the text are difficult to see in the figures, and I think that the original manuscript could be improved with a moderate revision. I report in what follows some comments and suggestions.

We thank the Reviewer for the thoughtful comments. In the following we answer each specific point (in blue).

Specific Comments:

1) All the variables assessed in this manuscript come from the atmospheric model component. I think that analysing the model bias of at least sea surface temperature (tos) could help complete the study. For example, the known Southern Ocean warm bias or North Atlantic cold bias in climate models are reported for the upper-ocean temperatures. The authors assess the 2m-air temperature (tas), which is fair, but it is not the same value as tos in the open ocean. Analysing the tos bias in climate models has an only sense for the coupled runs. But still, a different dataset (from ERA-interim) as observations should be used, and you could get new results. I would recommend looking at sea surface temperatures too.

Biases in sea surface temperature (SST) in the coupled models are nearly identical to those in the near-surface air temperature (SAT; compare the two Figures below with Figs. 1 and S2 in the submitted manuscript). Given that SSTs add no new information about temperature biases in the models (and are limited to the coupled models only, in contrast to the SAT), we will not include these figures, although we mention the results in Section 2 of the revised manuscript:

"Biases in SST (not shown) are very similar to those in SAT, which suggests SAT biases are dominated by the SST ones over the ocean."

[Figure]

Responses Fig. 1. (a) ERA5 sea surface temperature (SST; in °C), which are derived from the HadISST2 and OSTIA SST/SIC products (Hersbach et al., 2020). (b) Left and middle: Multi-model ensemble mean bias in SST (in °C) in the coupled models at low (left) and high (middle) resolutions. Biases are with respect to ERA5 (shown in a.). Stippling masks where at least four out of the five models agree on the anomaly sign. Right: Difference between the two resolutions. In all panels non-significant anomalies at the 5 % level (based on a two-tailed Student's t test) are masked white. The Equator is a dashed line in all the panels.

[Figure]

Responses Fig. 2. Mean bias in SST (in °C) in each individual coupled model at low (left) and high (middle) resolutions, and as the difference between the two (right). Biases are with respect to ERA5. Non-significant anomalies at the 5 % level are masked white. The Equator is a dashed line.

2) When discussing model biases regionally (in particular in sections 4.1, 5.1, 5.2), I think that a table similar to Table 2 (which I found very clear and helpful) could help to summarise the results. I suggest including additional tables in which the metrics RMSD and Bias are regionally computed.

Tables with regional biases have been included in the Supplementary Material (Tables S1–S4 in the revised manuscript) and the results are discussed throughout the revised manuscript. For example in Section 5.1:

"Increased resolution has a mixed effect on the SO warm bias and, although it seems to increase in the ensemble mean (Fig. 1), this varies substantially across models (Fig. S3 and Table S3): the CNRM-CM6-1 model experiences a reduction of a cold bias over the Weddell Sea up to about 4 °C; the EC-Earth3P warms along the Antarctic coast and its widespread SO warm bias persists at HR; the ECMWF-IFS model shows an increase of its temperature bias by about 1.5 °C on average and very strongly locally in the Weddell Sea by over 5 °C; the MPI-ESM1-2 shows a mean cooling over the SO of about 0.5 °C and becomes cold biased especially to the west of the Antarctic Peninsula; and the HadGEM3-GC31 model shows a reduction of its coastal cold bias, developing instead a more widespread warm bias with local values up to about 1–2 °C—although the cold bias over the Weddell Sea persists in the HadGEM3-GC31 eddy-rich model."

3) I may be wrong since I am not a native English speaker, but I find the concept 'improving/worsening the bias' (used several times along the manuscript) a little bit confusing. I would say that the model is improved/worsened by a reduction/increase in bias.

This has been rephrased throughout the text. Two examples from the revised manuscript:

"A reduction in the temperature and cloud biases in the eastern tropical oceans might reduce current uncertainty about climate sensitivity (Andrews et al., 2019)"

"This paper examines whether increased horizontal resolution reduces four well-known, long-standing climate biases in five global models developed within the PRIMAVERA project"

Minor Comments:

L182. Please re-order (Kato et al., 2018; Loeb et al., 2018).
Done.

L189-190. 'The statistical significance of the anomalies between models and observations is calculated based on a two-tailed Student's t-test at the 5 % level.' I do not understand this very well. For example, in figure 1, the stippling mask indicates areas where four out of five models agree on the sign of the bias. In the same figure, however, non-significant anomalies at the 5% level are masked white. What do you mean by 'anomalies' here? Are you referring to the bias itself? How did you consider the degrees of freedom? How many independent observations do you consider here? Please, specify.

Figures 1,2,4,5,6 include two sorts of significance analysis. The first one (stippling) masks areas where four out of the five models agree in the anoamy sign (where anomaly means the difference between a model, or the ensemble mean, and an observation). The second test (white mask) is a standard two-tailed Student's t test of whether the two samples (the model and the observation) have the same mean (null hypothesis), assuming they have different variances (Section 6.6: Test of the Mean; in Von Storch and Zwiers, 1999. *Statistical Analysis in Climate Research*. Cambridge Univ. Press). The sample size is the number of the years over which the mean is computed, indicated in Section 2.2. This point has been clarified in the revised manuscript:
"The statistical significance of the differences between models or the ensemble means and the observations is calculated for each variable based on a two-tailed Student's t test at the 5 % level, in which the null hypothesis is that the two samples (model and observations) have the same mean over the above-mentioned periods, assuming the two samples have different variances (von Storch and Zweirs, 1999). The associated non-significant values are masked in white in Figs. 1,2,4,5,6 and all the Supplementary Figures. An additional test is applied in Figs. 1,2,4,5,6 (shows as stippling) to measure the agreement in the difference's sign of the ensemble members with respect to observations.

L222. 'the bias is especially persistent': do you mean persistent across the models? Isn't the term 'persistent' related to time?
Rephrased: "At LR, the bias extends over the eastern tropical South Atlantic and South Pacific from the coast equatorward."

L240. In order of appearance, I would change the figure numbering and name Fig. 2 to current Fig. 4. Similarly, Fig. 5 would be Fig. 3.
Figures 2 and 3 are referred to in the Introduction, in the double ITCZ subseccion (1.1.2). Therefore, the figure order has not been changed.

L308. I guess that it is Fig. S6 instead of S8.
Corrected.

L317-318. Is the dry bias at SH mid-latitudes associated with the warm bias in the SO? I see the dry bias for each LR and HR model (Fig. S4) and even in the AMIP runs (Fig. S3).
This has been rephrased.
"The LR coupled models also present a dry bias at mid-latitudes (Fig. 2)."

L323. What do you mean by: 'an improvement of the cold bias in the Weddell Sea'?
It means the cold bias over the area gets reduced with increased resolution in the CNRM-CM6 model. This has been clarified as: "The CNRM-CM6-1 model experiences a reduction of a cold bias over the Weddell Sea up to about 4 °C".

L330-333. A table as table 2 with RMSD and mean bias computed regionally in the SO would help in this kind of statement.

Tables with regional biases have been included in the Supplementary Material (Tables S1–S4) and the results are discussed throughout the revised manuscript. For these particular lines, the revised text now is: "Increased resolution has a mixed effect on the SO warm bias and, although it seems to increase in the ensemble mean (Fig. 1), this varies substantially across models (Fig. S3 and Table S3): the CNRM-CM6-1 model experiences a reduction of a cold bias over the Weddell Sea up to about 4 °C; the EC-Earth3P warms along the Antarctic coast and its widespread SO warm bias persists at HR; the ECMWF-IFS model shows an increase of its temperature bias by about 1.5 °C on average and very strongly locally in the Weddell Sea by over 5 °C; the MPI-ESM1-2 shows a mean cooling over the SO of about 0.5 °C and becomes cold biased especially to the west of the Antarctic Peninsula; and the HadGEM3-GC31 model shows a reduction of its coastal cold bias, developing instead a more widespread warm bias with local values up to about 1–2 °C—although the cold bias over the Weddell Sea persists in the HadGEM3-GC31 eddy-rich model. In contrast to temperature, biases in cloud cover and net cloud radiative effect remain relatively unchanged between LR and HR (Figs. 4 and 5). The CNRM-CM6-1 shows a 1 % reduction in its mean cloud cover bias over the SO, while the ECMWF-IFS and MPI-ESM1-2 models show a 1–3 % increase over the SO (Table S3). Similarly, the ECMWF-IFS model shows a 1.5 Wm–2 mean reduction while the MPI-ESM1-2 model shows a 4 Wm–2 mean increase in their net cloud radiative effect biases over the SO (Figs. S6–S9)."

L333-337. Couldn't it also be the ocean model resolution or physics?

We have expanded this line to include the possibility that ocean resolution/physics have played a role in the Souther Ocean biases both in Section 5.1 (Southern Ocean bias) and Section 6 (Discussion and Conclusions):

"Given the small reduction in the cloud cover and net cloud radiative effect biases with increased resolution, the change in the temperature bias over the SO might be related to a change in the sensitivity of the HR coupled models to the similar cloud and radiation biases, or to development of further biases, for example, in the sea ice, mixed layer depth, air–sea heat fluxes, or the strength of the Antarctic Circumpolar Current (e.g., Roberts C.D. et al., 2018b). Some of these biases might, in turn, be linked to the disabling or not of the mesoscale eddy mixing at higher resolution (Roberts C.D. et al., 2018b), as discussed in Section 6."

"When additional model configurations are available, the benefit of bias reduction from increasing ocean resolution alone can be assessed. For the ECMWF-IFS model, increased ocean resolution from 1° to 0.25° reduces the North Atlantic, Arctic, and equatorial Pacific temperature biases but increases the SO warm biases (Roberts C.D., et al., 2018b). For the HadGEM3-GC31 model, increased ocean resolution up to an eddy-rich one (0.08°) improves the Gulf Stream

separation (Roberts M.J. et al., 2019) and representation (Moreno-Chamarro et al., 2021), although the eddy-rich resolution by itself has a modest impact on reducing surface temperature biases compared to the eddy-present (0.25°; Fig. S3 and Roberts M.J. et al., 2019). For the MPI-ESM1-2 model, the North Atlantic temperature and the Gulf Stream separation are also more realistic in an eddy-rich ocean (~0.1°) compared to the LR and HR versions used in our study (Gutjahr et al., 2019). These results thus suggest that an eddy-rich ocean resolution might be key to reducing North Atlantic and Southern Ocean temperature biases, which is consistent with previous studies (e.g., Mertens et al., 2014; Xu et al., 2014b). Particularly important for such biases might be the treatment of the mesoscale eddy mixing at the eddy-present resolution because mesoscale eddies become smaller at higher latitudes and are therefore not fully resolved at the eddy-present resolution (0.25°). Thus, for example, while the CNRM-CM6-1 (Voldoire et al., 2019b), EC-Earth3P (Haarsma et al., 2020), and MPI-ESM1-2 (Gutjahr et al., 2019) HR models respectively use a Smagorinsky scheme, and the Gent and Mcwilliams (1990) and K-profile parameterizations (KPP), the ECMWF-IFS (Roberts C.D., et al., 2018b) and HadGEM3-GC31 (Roberts M.J. et al., 2019) HR models switched off the Gent and Mcwilliams (1990) parametrization. Subtle differences in the model physics due to increased resolution might therefore exert a strong influence on model biases."

L365. 'All models but MPI-ESM1-2', not sure if also EC-Earth3P and CNRM-CM6-1. Again, here it would be helpful to have a table with the regional metrics.
This has been clarified: "Increased model resolution reduces the magnitude of the cold bias by about 1 °C on average (Table S4) and locally up to 2–3 °C in the ensemble mean (Fig. 1). There are, however, important differences across the ensemble members (Fig. S2). The EC-Earth and CNRM-CM6-1 HR models show relatively small local reductions of the cold bias by about 0.5–1 °C over the central subpolar North Atlantic. The lack of a clear improvement in these two HR models might be related to the unchanged ocean physics between the low and high resolutions (Section 2). The MPI-ESM1-2 shows no changes in the biases between resolutions over the subpolar North Atlantic but a strong cooling up to about 4 °C over the Nordic Seas, likely related to misrepresented local sea ice."

L373. 'In all the HR models, the cold bias over the subpolar North Atlantic is replaced by a warm bias'? I don't see it. Are you referring to the warm bias west of Greenland? If it is the case, it is not valid for MPI-ESM1-2.
This has been clarified: "On average at HR, the cold bias over the subpolar North Atlantic is replaced by a warm bias up to about 2–3 °C over the Labrador Sea (Fig. 1). The warming of the entire subpolar North Atlantic is, in fact, one of the most remarkable differences at increased resolution in the ensemble mean. The warming is especially prevalent in the NEMO models at the 0.25° resolution, in which the warm bias is likely related to a stronger oceanic heat transport in the North Atlantic and a reduced sea ice (Roberts M.J. et al., 2020b) than at lower resolutions, linked to a too strong ocean deep mixing in the Labrador Sea (Koenigk et al., 2021). In the

MPI-ESM1-2 models, by contrast, a warm bias is already present at LR and, together with the cold bias in the central North Atlantic bias, remains unchanged at HR (Fig. S3)."

L379. 'the cloud cover bias remains relatively unchanged' isn't easy to quantify from the figures. In the ensemble mean and in most of the individual models, changes in cloud cover between resolutions are of about ±5 % over the entire North Atlantic, with no evident changes in the pattern (Figs. 3, S5, and S6). This has been rephrased in the revised manuscript as "The change in the cloud cover bias in the ensemble means is relatively small, of about ±5 % over the entire North Atlantic, with no clear changes in the pattern (Figs. 3)." In addition we have included tables for the mean biases over each region to help quantify the changes (Tables S1–S4).

L398. 'tropics' instead of 'tropis'.
Corrected.

L416. How do you see that 'the Gulf Stream separation improves'? In HH, it appears a cold bias close to the coast.
This has been clarified. "For the HadGEM3-GC31 model, increased ocean resolution up to an eddy-rich one (0.08°) improves the Gulf Stream separation (Roberts M.J. et al., 2019) and representation (Moreno-Chamarro et al., 2021), although the eddy-rich resolution by itself has a modest impact on reducing surface temperature biases compared to the eddy-present (0.25°; Fig. S3 and Roberts M.J. et al., 2019)."

L432. In general, I would prefer the terms' eddy-parametrised/permitting/resolving' for the ocean resolution. But you are using 'eddy-parametrised/present/rich', so eddy-permitting here is not consistent. Besides, I think that you are referring to 'eddy-rich' instead of eddy-present' here.
We would like to keep the current eddy-parametrized, -present, -rich naming because it describes the models more accurately. For example, an "eddy-resolving" resolution of 1/12° does not actually resolve the typical ocean mesoscale eddies at high latitudes; a finer ocean resolution (1/20° or so) would be needed. The eddy-rich naming has become more frequent in recent literature (e.g., Malcolm M.J. et al., 2020). Nonetheless, the sentence the Reviewer points out has been corrected to follow this convention "Even though we acknowledge that our conclusions might be both model and region dependent, taken together, our analysis suggests that to remove model biases i) a refinement of the atmosphere resolution up to ~50-km alone might not always be sufficient, and ii) reaching eddy-rich ocean resolutions (1/12° or fine) might be needed. The increase in ocean resolution from eddy-parametrized (~100 km) to eddy-rich (~10 km) allows models to represent the first baroclinic Rossby radius and might therefore improve the representation of small-scale dynamical processes and then biases."

L455. I guess that is LL instead of LR.

The HadGEM3-GC31-LL configuration is referred to as LR (low-resolution) throughout the manuscript to simplify model naming. This is indicated in Section 2.1 and Table 1.

Figures and tables:
Table 2. Adding the global tos bias in the coupled runs could help determine the possible added value of increased ocean resolution.
As described before, biases in SST and SAT are very similar.

Figure 6. The stippling mask is missing in this figure.
The stippling mask has been modified so it does not look like horizontal lines.

---

## Author Comment (AC2)

**Referee: 2**

This is a nice study than analyses and compares important biases in PRIMAVERA models for high and low resolution. The ensemble is small but the authors spend sufficient time to discuss the differences between the models.

We thank the Reviewer for the thoughtful comments. In the following we answer each specific point (in blue).

Another issue is the separation between the impact of ocean and atmosphere resolution. The biases are often the result of coupled interactions and the present set of simulations make it difficult to separate the role of ocean and atmosphere resolution. I have sometimes the feeling that regarding the role of atmosphere or ocean resolution the authors jump too quickly to conclusions, thereby relying on other literature. I would prefer that they are a bit more cautious, if their conclusions cannot be without doubt supported by their own analyses. For instance I am not fully convinced that increasing atmosphere resolution is the main reason for the reduction of the warm bias in the upwelling regions as stated in the abstract. No clear analyses for that are provided.

Our conclusion on the benefit of increased atmosphere resolution derives from three models in particular, the ECMWF-IFS, HadGEM3-GC31 and MPI-ESM2-1. For the ECMWF-IFS model, C.D. Roberts et al. (2018b) show that the warm bias over the eastern tropical regions increases with increased ocean resolution between their MR (not included in our analysis) and LR resolutions but it is reduced with increased atmosphere resolution between the MR and the HR resolutions (see their Figure 3). For the HadGEM3-GC31, Roberts et al. (2019) compare model versions at different atmosphere and ocean resolutions, finding that the increase in atmosphere resolution from ~250 km to ~50 km reduces the warm bias over the eastern tropical regions, but the increase in ocean resolution from ~100 km to ~8 km shows no benefit there (see their Figure 7). Lastly, for the MPI-ESM1-2, only the resolution of the atmosphere increases between the LR (134 km) and HR (67 km) models in our analysis (Table 1 in the main manuscript) and therefore any bias reduction is due to increased atmosphere resolution. Bias reductions over the upwelling regions cannot be attributed to the increase in resolution in the other models (CNRM-CM6-1, EC-Earth3P) because i) the resolution increases in both the atmosphere and ocean between LR and HR, ii) there are no intermediate resolutions to test our hypothesis. This line has nonetheless been removed from the Abstract and has been clarified in the Discussion and Conclusions Section as "As for the increase in atmosphere resolution alone, it contributes to reducing the warm bias over the eastern tropical oceans in the ECMWF-IFS (Roberts C.D., et al., 2018b), HadGEM3-GC31 (Roberts M.J. et al., 2019) and MPI-ESM1-2 (this study) coupled models."

One of my main concerns is the abstract which to my opinion does not reflect very well the main conclusions of is even in contradiction. In the conclusions it is written: "On average (i.e., in the

ensemble mean), the warm eastern tropical ocean, the double ITCZ, and the cold North Atlantic improve at higher resolutions, while the SO warm bias worsens or persists in some models, and a new warm bias emerges in the Labrador Sea in all the models as a result of excessive Atlantic ocean heat transport (Roberts M.J. et al., 2020b) and excessive ocean deep mixing in the Labrador Sea in NEMO models at a 0.25° resolution (Koenigk et al., 2021)." This is a fair summary of the results. I do not see this reflected in the abstract, instead it speculates too much about the role of the atmosphere or ocean and the need of eddy rich ocean modelling. This can be discussed in the discussion section, but the abstract should be mainly limited to the results obtained from the analyses. I urge the authors to modify the abstract and make it more coherent with the main text.

We thank the Reviewer for raising this issue. We have reworked the Abstract entirely to remove any discussion and highlight the results:

"**Abstract.** We examine the influence of increased resolution on four long-standing biases using five different climate models developed within the PRIMAVERA project. The biases are the warm eastern tropical oceans, the double Intertropical Convergence Zone (ITCZ), the warm Southern Ocean, and the cold North Atlantic. Atmosphere resolution increases from ~100–200 km to ~25–50 km, and ocean resolution increases from ~1° (eddy-parametrized) to ~0.25° (eddy-present). For one model, ocean resolution also reaches 1/12° (eddy-rich). The ensemble mean and individual fully coupled general circulation models and their atmosphere-only versions are compared with satellite observations and the ERA5 reanalysis over the period 1980–2014. The four studied biases appear in all the low resolution coupled models to some extent, although the Southern Ocean warm bias is the least persistent across individual models. In the ensemble mean, increased resolution reduces the surface warm bias and the associated cloud cover and precipitation biases over the eastern tropical oceans, particularly over the tropical South Atlantic. Linked to this and to the improvement in the precipitation distribution over the western tropical Pacific, the double ITCZ bias is also reduced with increased resolution. The Southern Ocean warm bias increases or remains unchanged at higher resolution, with small reductions in the regional cloud cover and net cloud radiative effect biases. The North Atlantic cold bias is also reduced at higher resolution, albeit at the expense of a new warm bias that emerges in the Labrador Sea related to excessive ocean deep mixing in the region, especially in the ORCA025 ocean model. Overall, the impact of increased resolution on the surface temperature biases is model-dependent in the coupled models. In the atmosphere-only models, increased resolution leads to very modest or no reduction in the studied biases. Thus, both the coupled and atmosphere-only models still show large biases in tropical precipitation and cloud cover, and in mid-latitude zonal winds at higher resolutions, with little change in their global biases for temperature, precipitation, cloud cover, and net cloud radiative effect. Our analysis finds no clear reductions in the studied biases due to the increase in atmosphere resolution up to 25–50 km, in ocean resolution up to 0.25°, or in both. Our study thus adds to evidence that further improved model physics, tuning, and even finer resolutions might be necessary."

Specific comments:

L55. Here it is clear which biases are analyzed. This was not clear from the abstract.
The biases are now mentioned in the Abstract as well.

For the upwelling regions, I miss a discussion about the role of ocean mixing. For the Atlantic upwelling region see for instance: https://doi.org/10.1175/JCLI-D-19-0608.1
We have reshaped Section 1.1.1 to include this and other important references.

"This bias has long been related to the underestimation of the cloud cover, which leads to warming because of excessive shortwave radiation reaching the surface (e.g., Huang et al., 2007; Hu et al., 2008). The warm bias, in turn, weakens the lower tropospheric stability and thus hinders the formation of the stratocumulus deck, which contributes to sustaining the surface warm bias. Other mechanisms have been proposed to explain this bias, including too weak equatorial and alongshore winds weakening upwelling (e.g., Richter et al., 2012; Koseki et al., 2018; Goubanova et al., 2019; Voldoire et al., 2019a), biases in regional atmospheric moisture (Hourdin et al., 2015), too weak offshore transport by ocean mesoscale eddies, and the misrepresentation of the coastal current system (Xu et al., 2014) or vertical mixing in the upper ocean (e.g., Hazeleger and Haarsma, 2005; Exarchou et al., 2018; Deppenmeier et al. 2020)."

L 180. I was surprised that ERA-Interim analyses were used and not the more recent ERA5.
ERA5 is now used for ERA-Interim. The results remain the same, which is mentioned in the revised manuscript: "Biases in SAT and zonal winds with respect to the ERA-Interim reanalysis (Dee et al., 2011) are very similar to those with respect to ERA5 (not shown)."

L 230. I do not think that on the basis of one eddy rich model and no dedicated analyses between increase of ocean and atmosphere resolution, you can make that statement.
As previously mentioned, this is supported by the analysis in Roberts M.J. et al., 2019, in which model versions of the HadGEM3-GC31 at different atmosphere and ocean resolutions are compared. The authors find that increased atmosphere resolution tends to lead to reduced temperature biases over the upwelling regions, but increased ocean resolution offers no major benefit (see their Fig. 7).

L 376. Why not mention this in the abstract?
This is now included in the Abstract.

L 420. This statement seems to contradict for instance with L364 and L376. The reduction of the cold bias in the sub-polar gyre in the North Atlantic is one of the strongest signals between LR and HR. Also, the reduction of the double ITCZ bias in the Pacific is a clear signal that the

authors partly attribute to the increase in the ocean model resolution (L290). So, this statement is not backed up by the authors own analyses. This statement is then lifted to the abstract, where it should be removed. It can be discussed in the discussion section with a reference to the statement at L370 where eddy-rich ocean models improve the Gulfstream separation and reduce the warm bias near the coast.

The line has been removed entirely.

Typo's

L216: … small impact… The structure of the sentence suggests positive is missing between small and impact.

Increasing resolution does not necessarily reduce global biases (positive impact); it can also increase the bias (for example, as for the surface temperature and precipitation in the ensemble mean). Nonetheless, increased resolution has most of the times a small impact in the global bias, as stated in the manuscript.

L303:  Referring only to Fig. 1 and not also to Fig. S2 reads strange in the first sentence.

Corrected.

L348: ... compared to...

Corrected.

L425 ..helps to reduce..

Rephrased.